EMBO
Molecular Medicine

# ZEB1-mediated melanoma cell plasticity enhances resistance to MAPK inhibitors

Geoffrey Richard[1,2,3,4,5], Stéphane Dalle[1,2,3,4,5,6], Marie-Ambre Monet[1,2,3,4,5], Maud Ligier[1,2,3,4,5], Amélie Boespflug[1,2,3,4,5,6], Roxane M Pommier[1,2,3,4,5], Arnaud de la Fouchardière[1,2,3,4,5,7], Marie Perier-Muzet[1,2,3,4,5,6], Lauriane Depaepe[8], Romain Barnault[1,2,3,4,5], Garance Tondeur[8], Stéphane Ansieau[1,2,3,4,5], Emilie Thomas[9], Corine Bertolotto[10,11,12], Robert Ballotti[10,11,12], Samia Mourah[13], Maxime Battistella[14], Céleste Lebbé[15,16], Luc Thomas[1,2,3,4,5,6], Alain Puisieux[1,2,3,4,5,17,*] & Julie Caramel[1,2,3,4,5,**]

## Abstract

Targeted therapies with MAPK inhibitors (MAPKi) are faced with severe problems of resistance in *BRAF*-mutant melanoma. In parallel to the acquisition of genetic mutations, melanoma cells may also adapt to the drugs through phenotype switching. The ZEB1 transcription factor, a known inducer of EMT and invasiveness, is now considered as a genuine oncogenic factor required for tumor initiation, cancer cell plasticity, and drug resistance in carcinomas. Here, we show that high levels of ZEB1 expression are associated with inherent resistance to MAPKi in *BRAF^V600*-mutated cell lines and tumors. ZEB1 levels are also elevated in melanoma cells with acquired resistance and in biopsies from patients relapsing while under treatment. *ZEB1* overexpression is sufficient to drive the emergence of resistance to MAPKi by promoting a reversible transition toward a MITF^low/p75^high stem-like and tumorigenic phenotype. ZEB1 inhibition promotes cell differentiation, prevents tumorigenic growth *in vivo*, sensitizes naive melanoma cells to MAPKi, and induces cell death in resistant cells. Overall, our results demonstrate that ZEB1 is a major driver of melanoma cell plasticity, driving drug adaptation and phenotypic resistance to MAPKi.

**Keywords** EMT; MAPK; melanoma; resistance; targeted therapy

**Subject Categories** Cancer; Skin

## Introduction

The recent emergence of targeted therapies directed against components of the mitogen-activated protein kinase (MAPK) pathway has led to unprecedented clinical benefits for the treatment of metastatic melanoma. Approximately 50% of melanomas exhibit a *BRAF^V600E* mutation that can be targeted with specific inhibitors, namely vemurafenib (or PLX4032) or dabrafenib (Chapman *et al*, 2011). Unfortunately, resistance to these BRAF inhibitors (BRAFi) invariably develops in patients after only a few months, through various mechanisms that generally lead to the reactivation of the BRAF-MEK-ERK pathway or to the activation of the PI3K-AKT survival pathway (Lito *et al*, 2013; Holderfield *et al*, 2014; Van Allen *et al*, 2014). Fifty percent of patients displaying mutated *BRAF* melanomas also develop intrinsic/innate resistance at an early stage during treatment. The combined administration of BRAFi and MEK inhibitors (MEKi), such as trametinib or cobimetinib, has been shown to

1  Cancer Research Center of Lyon, INSERM U1052, Lyon, France
2  Cancer Research Center of Lyon, CNRS UMR 5286, Lyon, France
3  Université de Lyon, Lyon, France
4  ISPB, Université Lyon 1, Lyon, France
5  Centre Léon Bérard, Lyon, France
6  Dermatology Unit, Hospices Civils de Lyon, CH Lyon Sud, Pierre Bénite Cedex, France
7  Department of Biopathology, Centre Léon Bérard, Lyon, France
8  Department of Biopathology, Hospices Civils de Lyon, CH Lyon Sud, Pierre-Bénite Cedex, France
9  Fondation Synergie Lyon Cancer, Centre Léon Bérard, Lyon, France
10 INSERM U1065, Equipe 1, Biologie et pathologies des mélanocytes: de la pigmentation cutanée au mélanome, Equipe labellisée Ligue 2013, Centre Méditerranéen de Médecine Moléculaire, Nice, France
11 Université de Nice Sophia-Antipolis, UFR Médecine, Nice, France
12 CHU Nice, Service de Dermatologie, Nice, France
13 APHP, INSERM U976, Saint Louis Hospital Pharmacology-Genetic Laboratory Paris, Paris, France
14 Department of Pathology, INSERM U1165, Université Paris Diderot, AP-HP, Hôpital Saint-Louis, Paris, France
15 Department of Dermatology, APHP, Saint Louis Hospital, Paris, France
16 INSERM U976, University Paris 7 Diderot, Paris, France
17 Institut Universitaire de France, Paris, France
   *Corresponding author. Tel: +33 478782667; E-mail: alain.puisieux@lyon.unicancer.fr
   **Corresponding author. Tel: +33 426556740; E-mail: julie.caramel@lyon.unicancer.fr

extend median progression-free survival, from 7.3 months in the vemurafenib monotherapy group to 11.4 months in the combination therapy group (Robert *et al*, 2015). However, combination therapy does not prevent the appearance of acquired resistance (Flaherty *et al*, 2012), through genetic mechanisms resembling those described during monotherapy (Long *et al*, 2014; Moriceau *et al*, 2015). Moreover, no clear mutational mechanism is found in up to forty percent of resistant melanomas (Hugo *et al*, 2015), indicating that transcriptomic or epigenetic alterations may underly acquired MAPK inhibitor (MAPKi) resistance. Therefore, identification of non-genomic mechanisms may lead to the design of more efficient combination therapies.

We and others have recently demonstrated the role of epithelial–mesenchymal transition-inducing transcription factors (EMT-TFs) in the development of melanoma (Shirley *et al*, 2012; Caramel *et al*, 2013; Denecker *et al*, 2014; Tulchinsky *et al*, 2014). EMT is a reversible embryonic process that is often aberrantly reactivated during the progression of carcinomas (Trimboli *et al*, 2008; Morel *et al*, 2012), where it promotes invasion and metastatic dissemination (Thiery *et al*, 2009). More recently, we and others have shown that EMT-TFs of the ZEB, SNAIL, and TWIST families act as genuine oncogenic factors in epithelial cells, promoting cell transformation, stemness (Mani *et al*, 2008; Morel *et al*, 2008) and carcinoma initiation *in vivo* (Liu *et al*, 2014; Puisieux *et al*, 2014; Beck *et al*, 2015). ZEB1 is the main regulator of breast cancer cell plasticity enabling the reversible conversion of non-CSCs (cancer stem cells) into CSCs (Chaffer *et al*, 2013). EMT commitment has also been shown to promote resistance to treatment, establishing a link with the resistant phenotype of CSCs (Polyak & Weinberg, 2009; Singh & Settleman, 2010; Mallini *et al*, 2014; Tan *et al*, 2014). However, the epithelial phenotype was instead associated with resistance to cisplatin in ovarian carcinoma, therefore highlighting cancer type dependency and drug specificity (Tan *et al*, 2014; Miow *et al*, 2015).

Conversely to what has been reported for carcinoma, not all EMT-TFs exhibit oncogenic functions in melanoma. Indeed, ZEB2 and SNAIL2 are expressed in normal adult melanocytes, and a switch in EMT-TFs expression, characterized by a loss of ZEB2 and SNAIL2 and an upregulation of ZEB1 and TWIST1, occurs during melanoma progression (Shirley *et al*, 2012; Caramel *et al*, 2013; Denecker *et al*, 2014; Tulchinsky *et al*, 2014). This reversible switch is regulated by the MAPK pathway, at least in part through the AP1 JUN-FRA1 transcriptional complex, and represents a major risk factor for a poor outcome in melanoma patients.

We herein wondered whether high ZEB1/TWIST1 expression may be associated with the resistance to MAPKi in melanoma, with the final aim of testing whether targeting these factors in combination with MAPKi could prevent the emergence of resistance. We uncovered that high ZEB1 expression levels were associated with inherent resistance to MAPKi in $BRAF^{V600}$-mutated cell lines and tumors. Moreover, ZEB1 expression level was increased in melanoma cell lines with acquired resistance to BRAFi and in biopsies from patients relapsing while under BRAFi treatment. We further demonstrated that *ZEB1* overexpression in melanoma cell lines triggered the emergence of resistance to MAPKi by promoting the reversible conversion of a MITF$^{high}$/p75$^{low}$ differentiated state into a MITF$^{low}$/p75$^{high}$ stem-like and tumorigenic state. Consequently, the inhibition of ZEB1 sensitized naive melanoma cells to BRAFi, prevented the emergence of resistance following chronic exposure to BRAFi *in vitro*, and induced cell death in resistant melanoma cells. Collectively, these data highlight the role of ZEB1 as a major driver of phenotype switching in melanoma cells, providing them with a resistance to MAPKi.

# Results

## High levels of ZEB1 expression are correlated with low MITF levels and are associated with inherent resistance to MAPKi in $BRAF^{V600}$-mutated melanoma cell lines

The microphthalmia-associated transcription factor (MITF) is the master regulator of melanocyte development, and MITF expression levels tightly regulate and control the phenotype of melanoma cells (Hoek & Goding, 2010). Indeed, according to the MITF rheostat model, MITF regulates the transition from a differentiated, cell cycle-arrested phenotype (MITF$^{high}$) to a proliferative phenotype (intermediate MITF) and then to a quiescent stem-like phenotype (MITF$^{low}$). Interestingly, previous expression microarray profiling of murine immortalized melan-a cells revealed that the ectopic expression of *ZEB1* or *TWIST1* induces the downregulation of *MITF* (Caramel *et al*, 2013). Taking these findings into consideration, we analyzed the crosstalk between EMT-TFs and MITF in human melanoma cells. We found a strong inverse correlation between the mRNA expression of *ZEB1* and *MITF* in melanoma cell lines from the Cancer Cell Line Encyclopedia (CCLE), regardless of their *BRAF/NRAS* mutational status ($n = 61$, $P = 4.08E-11$; Fig 1A). As expected from our previous results (Caramel *et al*, 2013), the level of *ZEB2* expression was inversely correlated with *ZEB1* and thus positively associated with *MITF* (Appendix Fig S1). In contrast, the expression of *TWIST1* showed no significant correlation with that of *MITF*

**Figure 1. High levels of ZEB1 expression are correlated with low MITF levels and are associated with inherent resistance to MAPKi in $BRAF^{V600}$-mutated melanoma cell lines.**

A   *MITF* mRNA expression according to ZEB1 expression levels in 61 melanoma cell lines available through the CCLE (Pearson correlation test).
B   ZEB1, ZEB2, TWIST1, and MITF expression in a panel of $BRAF^{V600}$-mutated melanoma cells assessed by Western blot. GLO and C-09.10 cells are patient-derived short-term cultures. Actin was used as a loading control.
C   Quantitative PCR analyses of *ZEB1*, *ZEB2*, *TWIST1*, and *MITF* in the same panel of cell lines. mRNA expression levels are represented relative to C-09.10 cells, in which the levels were arbitrarily fixed at 1 ($n = 3$, mean ± SD). The dotted line separates ZEB1$^{high}$ (left) and ZEB1$^{low}$ (right) cell lines.
D   *ZEB1*, *ZEB2*, *TWIST1*, and *MITF* mRNA expression according to the IC$_{50}$ of the drug (μM) administered (BRAFi/MEKi), in melanoma cell lines from the CCLE ($n = 28$) (Tukey box plot, Student's *t*-test). High *ZEB1*, low *ZEB2*, and low *MITF* expression levels were correlated with BRAFi (PLX4720) and MEKi (AZD6244) resistance. PLX4720 is an analog of PLX4032.
E   IC$_{50}$ values of PLX4032 (μM) in the panel of $BRAF^{V600}$ melanoma cells as determined by ATP assay ($n = 3$, mean ± SD). For SKMEL24 and WM793, IC$_{50}$ was > 8 μM.

Source data are available online for this figure.

                                                                 

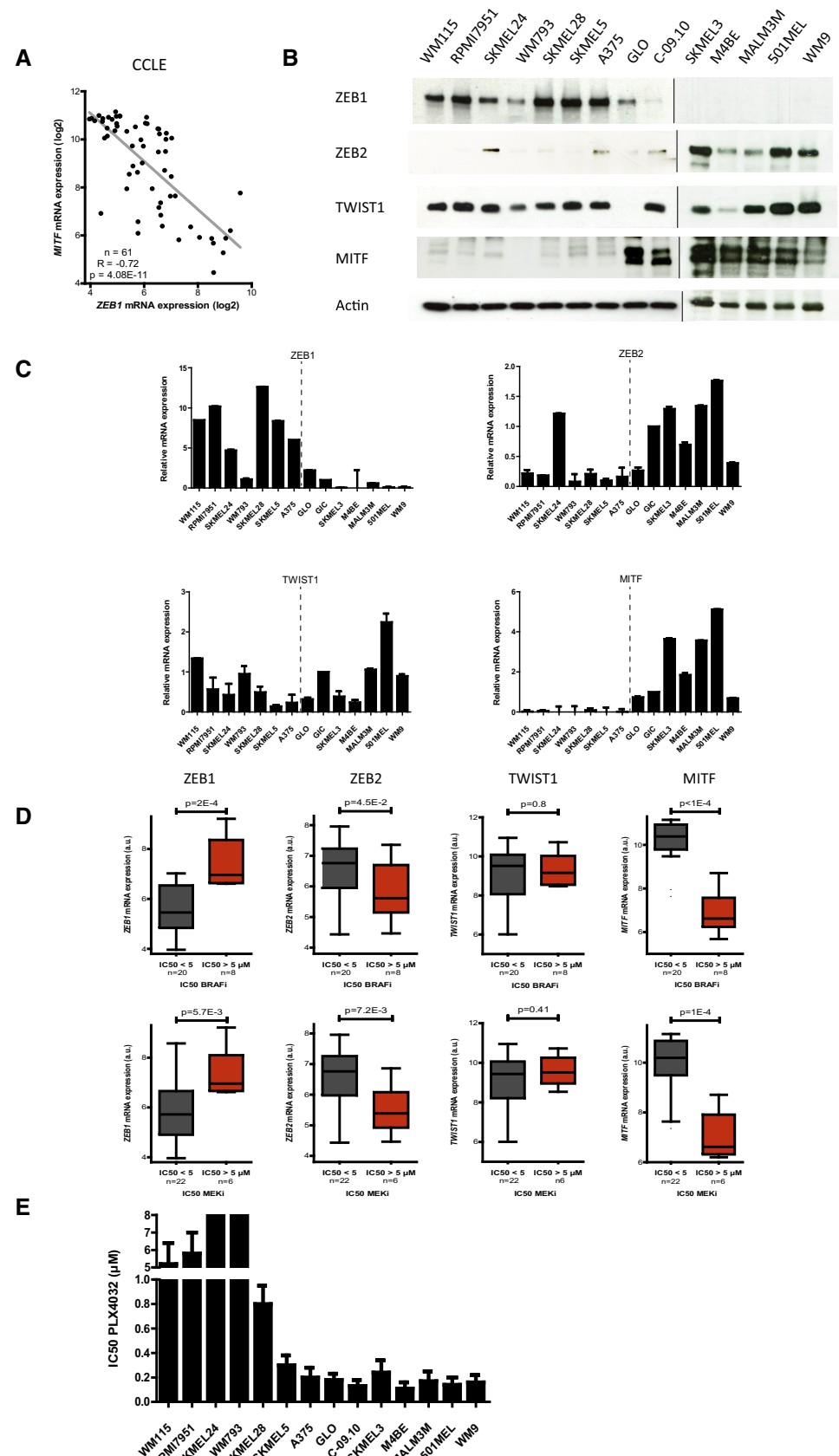

Figure 1.

(Appendix Fig S1). We then confirmed these results by conducting quantitative PCR (Q-PCR) and Western blot analyses in a panel of 14 $BRAF^{V600}$-mutated human melanoma cell lines, including two short-term cultures established from patients with melanomas displaying similar mutations (GLO and C-09.10; Fig 1B and C). We observed an inverse correlation between the levels of ZEB1 and those of ZEB2 and MITF, while TWIST1 protein levels were generally high in all of these cell lines and were not correlated with MITF (Fig 1B and C).

An increase in the expression of MITF was previously reported to contribute to tumor progression and resistance to BRAFi in a subset of melanomas (Johannessen et al, 2013), whereas low levels of MITF expression were shown to predict intrinsic MAPKi resistance (Konieczkowski et al, 2014; Muller et al, 2014), highlighting the dual function of this factor. We thus investigated whether the sensitivity of 28 $BRAF^{V600}$-mutated melanoma cell lines from the CCLE to BRAFi and MEKi was correlated with their EMT-TF/MITF expression profiles. We observed a significant inverse correlation between the level of $ZEB1$ mRNA and sensitivity to the BRAFi PLX4720 ($n = 28$, $P = 2E-4$), with BRAFi-insensitive cell lines displaying the highest $ZEB1$ expression levels (Fig 1D, Appendix Fig S1). A similar correlation was observed for $ZEB1$ and inherent resistance to the MEKi AZD6244 ($P = 5.7E-3$; Fig 1D, Appendix Fig S1). In contrast, high levels of $ZEB2$ expression were correlated with low levels of $ZEB1$ expression and with a higher sensitivity to BRAFi and MEKi (Fig 1D, Appendix Fig S1). No correlation with $TWIST1$ was observed (Fig 1D, Appendix Fig S1), indicating that not all EMT-TFs are implicated in the regulation of MAPKi sensitivity in melanomas. As previously suggested (Konieczkowski et al, 2014; Muller et al, 2014), low $MITF$ levels were associated with intrinsic resistance to MAPKi in these cell lines. We then validated these findings in our panel of $BRAF^{V600}$-mutated melanoma cell lines, by determining the $IC_{50}$ of PLX4032. To do so, we performed ATP assays, in which melanoma cells were treated with this drug, at a dose ranging from 1 nM to 10 μM, for 72 h. The $IC_{50}$ for PLX4032 was generally higher in the ZEB1$^{high}$/MITF$^{low}$ cell lines, compared to the ZEB1$^{low}$/MITF$^{high}$ cell lines (Fig 1E). Overall, these in silico and in vitro data demonstrate that cell lines intrinsically resistant to MAPKi exhibit a ZEB1$^{high}$/MITF$^{low}$ profile.

**High ZEB1 and low MITF levels are associated with inherent resistance to MAPKi in $BRAF^{V600}$-mutated melanoma tumors in patients**

We then investigated the relevance of these in vitro observations in human melanoma samples. The correlation between high $ZEB1$ and low $MITF$ expression was confirmed in a collection of 467 primary and metastatic melanomas from The Cancer Genome Atlas (TCGA; Cerami et al, 2012; Gao et al, 2013; The Cancer Genome Atlas Network, 2015) ($P < 2.2E-16$; Fig 2A). Interestingly, in this cohort, $ZEB1$ expression was higher in $BRAF^{V600}$ or $NRAS^{Q61R}$-mutated melanomas compared to $BRAF/NRAS$ WT tumors (Appendix Fig S2), which corroborates the involvement of the MAPK pathway in the regulation of ZEB1. To determine whether the levels of ZEB1 and MITF were predictive of the patients' response to MAPKi, we performed immunohistochemical staining for ZEB1, MITF but also TWIST1 on a cohort of 70 human $BRAF^{V600}$ melanoma samples from patients whose response to the treatment was known. Thirty patients presented a primary resistance (initial non-responders), and 40 were initial responders but relapsed during their treatment with MAPKi (developing acquired resistance). Sixteen of those patients received combined treatment with the MEK inhibitor cobimetinib. In some cases, ZEB1 staining was observed as a gradient from superficial to deep sites (Fig 2B), as previously described (Caramel et al, 2013). Interestingly, in other cases, ZEB1 was not only detected in the invasive front but also in the bulk of the tumor (Fig 2C), suggesting that in addition to its role in promoting tumor invasion, it may also be implicated in tumor development. Regarding MITF staining, approximately half of the samples presented a strong and homogeneous expression of MITF, while heterogeneous staining was observed in the second half, with clones exhibiting a loss of or low levels of the MITF protein. In most of those cases, a lower level of MITF was found in a gradient from superficial to deep sites and was correlated with higher levels of ZEB1 (Fig 2B). Of note, the faint ZEB1/MITF correlation obtained from the TCGA compared to that observed in cell lines may be due to ZEB1/MITF intra-tumoral heterogeneity. Interestingly, TWIST1 was also detected in most of the ZEB1-positive cases, although the intensity and percentage of positive cells were generally lower for TWIST1 compared to ZEB1 (Fig EV1A). Once again, the TWIST1 levels within the tumors were not correlated with MITF levels.

In order to correlate the variation in ZEB1 levels with the response to treatment, a ZEB1 staining score was defined based on the intensity and percentage of positive cells. The samples were divided into three groups (Fig 2C): "ZEB1$^{high}$" was defined as tumors with 80–100% positive cells showing a strong staining intensity, "ZEB1$^{int}$" (intermediate) included samples with 40–60% positivity with a moderate intensity and 60–80% positivity with a low intensity, whereas "ZEB1$^{low}$" corresponded to samples with fewer than 40% positive cells with a low to moderate intensity. Interestingly, most ZEB1$^{high}$ melanoma samples were in the primary

**Figure 2. High ZEB1 and low MITF levels are associated with inherent resistance to MAPKi in $BRAF^{V600}$-mutated melanoma tumors.**

A    $MITF$ mRNA expression levels according to ZEB1 expression in 467 melanoma tumors from the TCGA data set (Pearson correlation test).

B    Representative pictures of ZEB1 and MITF immunostaining in primary melanomas. Scale bar = 40 μm. The aberrant activation of ZEB1 in melanomas is correlated with a MITF$^{low}$ phenotype.

C    Representative pictures of ZEB1 immunostaining in $BRAF^{V600}$ tumors from patients, classified into ZEB1$^{high}$, ZEB1$^{int}$, and ZEB1$^{low}$ subgroups based on the intensity of ZEB1 staining and on the percentage of cells positive for ZEB1. Scale bar = 80 μm.

D    Pie charts representing the distribution of ZEB1 alone (upper part), or ZEB1 and TWIST1 (lower part) immunohistochemical staining in tumors according to their initial response to vemurafenib ± cobimetinib treatment. ZEB1 ± TWIST1 levels are higher in MAPKi primary resistant melanomas (initial non-responders) compared to tumors that initially respond to treatment ($n = 70$, Fisher's exact test).

E    Representative pictures of ZEB1 and MITF immunostainings, before and after vemurafenib treatment, in the tumor from patient 1, exhibiting primary resistance to BRAFi. Scale bar = 80 μm. Right: Magnification of MITF$^{high}$ and MITF$^{low}$ clones in the resistant tumor under treatment. Arrows indicate endothelial and stromal cells that also show positive staining for ZEB1, besides tumor cells.

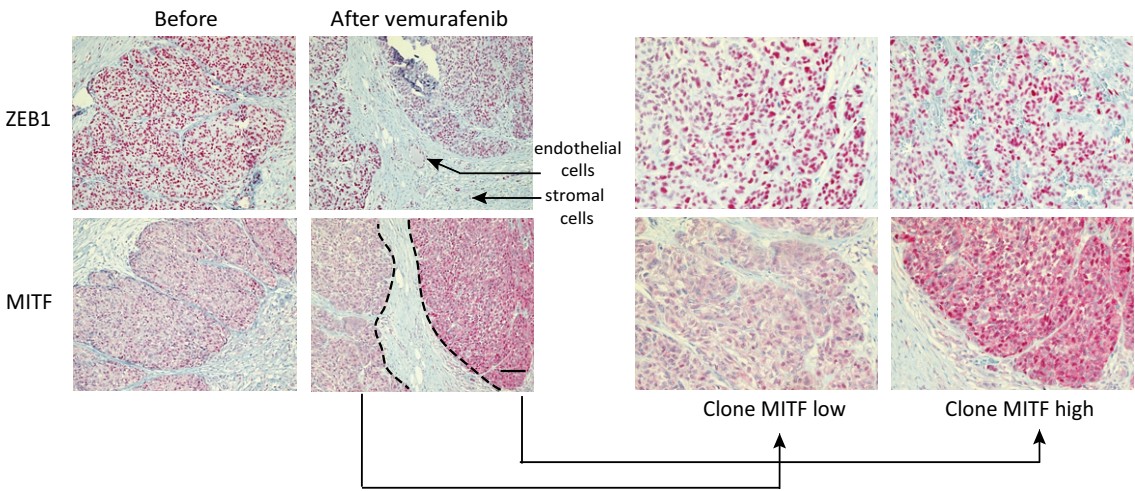

Figure 2.

resistance group (Fig 2D). Thirty percent of primary resistant melanomas exhibited high levels of endogenous ZEB1, compared to only 7.5% of the initially responding tumors ($P = 2.27E-2$; Fig 2D). Moreover, the samples with high levels of ZEB1 exhibited low levels of MITF (patient 1, before treatment, Fig 2E). Interestingly, a significant proportion of ZEB1-negative melanoma samples from patients with primary resistance displayed strong TWIST1 staining. Collectively, 50% of primary resistant melanomas exhibited a strong staining for ZEB1 and/or TWIST1. Thus, initial high levels of endogenous ZEB1 and/or TWIST1 were associated with primary resistance to treatment ($P = 3E-4$; Fig 2D). Moreover, biopsies conducted after vemurafenib treatment confirmed that a high level of ZEB1 was maintained in three of three sample pairs (patient 1, after vemurafenib, Fig 2E). Altogether, an inverse correlation was observed between ZEB1 and MITF levels in $BRAF^{V600}$-mutated melanoma at the intra-tumoral level, and this ZEB1$^{high}$/MITF$^{low}$ profile showed a trend toward intrinsic resistance to MAPKi.

## ZEB1 expression is activated in melanoma cell lines with acquired resistance to BRAFi and in biopsies from patients relapsing while under treatment

To investigate the putative role of ZEB1 in the development of acquired resistance to BRAFi, we established two lines of BRAFi-resistant melanoma cells. A375 and SKMEL5 human $BRAF^{V600}$ melanoma cells were treated with increasing doses of PLX4032 for 8 weeks to generate resistant cell lines, subsequently referred to as A375-R and SKMEL5-R. These cells exhibited a 10-fold increase in their IC$_{50}$ value for PLX4032 compared with the sensitive parental cells (Fig 3A). The resistant cells displayed a strong increase in their levels of ZEB1 protein and mRNA compared to their parental counterparts (Fig 3B and C). The protein levels of the FRA1 transcription factor, a known inducer of ZEB1 in melanomas, also increased, whereas TWIST1 was not affected. It is worth noting that MITF mRNA levels were lost in A375-R but increased in SKMEL5-R (Fig 3C). We also established two BRAFi-resistant short-term cultures from ascites of $BRAF^{V600}$-mutated patients, which had acquired a resistance to vemurafenib (GOKA and ESP). These patients initially responded to BRAFi but became resistant within a few months after the onset of the treatment. These BRAFi-resistant cells exhibited a high level of ZEB1 expression equivalent to that of resistant A375-R and SKMEL5-R cells (Fig 3B and C). Immunohistochemical analyses of a metastatic melanoma sample from the patient, from whom the ESP cells were established, revealed low ZEB1 and TWIST1 protein levels before treatment (Fig EV1C), confirming the increase in ZEB1 expression upon acquisition of resistance to vemurafenib. The level of MITF mRNA expression was low in GOKA but remained elevated in resistant ESP cells, while ZEB1 was high in all of these in vitro resistant models (Fig 3C).

To assess the relevance of our findings in a physiologically more relevant setting, we investigated the levels of ZEB1 and TWIST1 expression in tumors from a cohort of patients before and after vemurafenib treatment. In five out of eight matched pre-treatment and post-relapse sample pairs of acquired resistance, either ZEB1 proteins appeared or their levels had further increased (Fig 3D). In the samples obtained from an initially responding patient, both ZEB1 and TWIST1 were present at low levels and then increased significantly in the relapsed tumor after treatment, in terms of both

the intensity and percentage of positive cells, which changed from 10% to 80% (patient 4, Figs 3D and EV1B). We observed a decrease in MITF levels after treatment in two out of eight patients (some clones of patient 1, Fig 2E), while these levels were maintained (patient 2, Fig 3D) or increased (patients 3 and 4, Fig 3D) in most melanoma relapse samples after vemurafenib treatment. High levels of ZEB1 protein could be detected in both MITF$^{high}$ and MITF$^{low}$ clones after vemurafenib treatment, even within the same tumor (patient 1, Fig 2E). Overall, our findings indicate that increased ZEB1 expression is a common event in acquiring resistance to vemurafenib but is not necessarily associated with a loss in MITF expression in vitro or in tumors from patients. These observations suggest that the function of ZEB1 in MAPKi resistance may be mediated by MITF-dependent and MITF-independent mechanisms. This prompted us to investigate how ZEB1 promotes intrinsic or acquired resistance to MAPKi in $BRAF^{V600}$-mutant melanoma.

## ZEB1 overexpression promotes stemness properties, tumorigenic capacity, and resistance to MAPKi

To further investigate the functions of ZEB1 in the modulation of melanoma cell plasticity and resistance to MAPKi, we selected two ZEB1$^{high}$/MITF$^{low}$ human melanoma cell lines (A375 and SKMEL5), two ZEB1$^{low}$/MITF$^{high}$ short-term cultures (C-09.10 and GLO), and one ZEB1$^{low}$/MITF$^{high}$ cell line (501MEL) (Fig 1B). ZEB1 was ectopically expressed in these different models by infecting cells with ZEB1-expressing retroviruses. First, in A375 and SKMEL5, although already expressing ZEB1, its expression level could still be significantly increased (Fig 4A, Appendix Fig S3A) and proliferation was not affected. ZEB1 ectopic expression triggered a downregulation of MITF (Fig 4B and Appendix Fig S3B) as well as upregulation of ZEB2 (Appendix Fig S4A). In addition to low levels of MITF, different markers, including ABCB5 and JARID1B (Schatton et al, 2008; Roesch et al, 2010), have been associated with the generation of melanoma-initiating cells. Therefore, we analyzed the expression of these two factors by Q-PCR and uncovered an activation following the ectopic expression of ZEB1 (Fig 4B). Analyses of the expression of the neural crest cell marker p75/CD271, another melanoma-initiating cell marker (Boiko et al, 2010; Civenni et al, 2011), by Western blot and Q-PCR, consistently revealed its upregulation in A375 and SKMEL5 cells that overexpressed ZEB1 (Fig 4A and B, Appendix Fig S3B). Moreover, p75 mRNA expression was positively correlated with ZEB1 in melanoma samples from the TCGA ($R = 0.29$; $P = 1.16E-10$; Appendix Fig S5). Moreover, expression levels of some invasion markers, such as Vimentin, SPARC, or MMP1, were slightly induced in ZEB1-overexpressing cells, while AXL and WNT5A levels were not modified (Appendix Fig S4B).

We then analyzed the oncogenic functions of ZEB1 in melanoma cells. ZEB1 overexpression promoted the growth of A375 and SKMEL5 cells in a semi-solid medium (Fig 4C, Appendix Fig S3C). A 50% increase in the number of colonies growing from cells overexpressing ZEB1 was observed compared to control cells, and was associated with a concomitant increase in the size of the colonies. Xenograft experiments in nude mice were performed with control or ZEB1-overexpressing A375 cells and revealed a significant increase in tumor growth in the latter case (Fig 4D). Next, we examined whether the ectopic expression of ZEB1 could enhance adaptive resistance to BRAFi. A375 and SKMEL5 cells were treated with

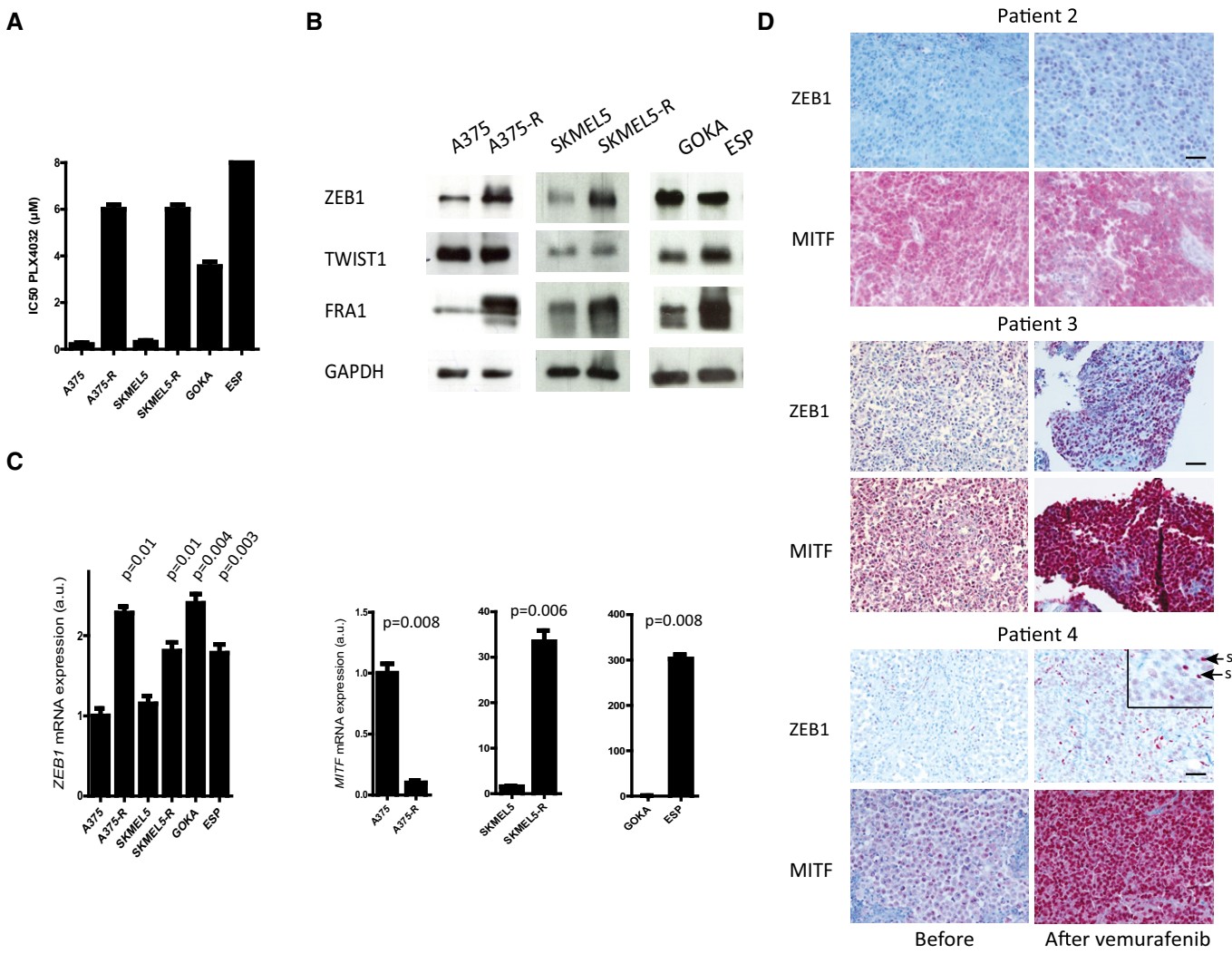

**Figure 3.  ZEB1 expression is activated in *BRAF^V600*-mutated melanoma cell lines with acquired resistance to BRAFi and in biopsies from patients relapsing while under treatment.**

A   PLX4032 IC$_{50}$ (μM) of sensitive A375 and SKMEL5 and resistant (A375-R, SKMEL5-R) cell lines, as well as of GOKA and ESP cells, two BRAFi-resistant patient-derived short-term cultures, as determined by ATP assay ($n = 3$, mean ± SD). For ESP, IC$_{50}$ was > 8 μM.

B   Western blot analyses of ZEB1, TWIST1, and FRA1 in A375-R and SKMEL5-R versus the parental naive cells, and in GOKA and ESP cells. GAPDH was used as a loading control.

C   Quantitative PCR analyses of *ZEB1* and *MITF* in A375-R and SKMEL5-R versus the parental naive cells, and in GOKA and ESP cells. mRNA expression levels are represented as arbitrary units (a.u.). Statistical difference relative to sensitive A375 cells is shown ($n = 3$, mean ± SD, Student's *t*-test).

D   Representative pictures of ZEB1 and MITF immunostainings in tumors from patients 2, 3, and 4, before and after vemurafenib treatment. Scale bars = 40 μm. For ZEB1 staining in patient 4, the inset shows a magnification. Arrows point at stromal cells (s). All other cells positive for ZEB1 are tumor cells.

Source data are available online for this figure.

PLX4032 at the IC$_{50}$ dose of the control cells (150 or 300 nM, respectively), which led to the inhibition of the MAPK pathway, as assessed by a reduction in the level of ERK phosphorylation (Fig 4E). The levels of ZEB1 and TWIST1 decreased upon treatment with PLX4032 in the control cells, whereas the level of MITF increased, which is consistent with the role of the MAPK pathway in the modulation of the two EMT-TFs. Furthermore, the level of ZEB1 remained higher in *ZEB1*-overexpressing cells compared with control cells, upon treatment with PLX4032 (Fig 4E). We then analyzed the surface distribution of p75 in the different cell populations by flow cytometry. ZEB1-mediated conversion into a p75^high phenotype was significantly potentiated in A375 (Fig 4F) and SKMEL5 cells (Appendix Fig S3D) following their treatment with PLX4032 for 10 days, indicating an early adaptation to the drug. These results suggested that ZEB1 induces a reprogramming of the cells upon BRAF inhibition and that ZEB1^high cells are prone to induce a p75^high stem-like phenotype in response to treatment with PLX4032.

To investigate the role of p75 in the ZEB1-mediated phenotype, siRNA experiments against *p75* were performed in A375 cells

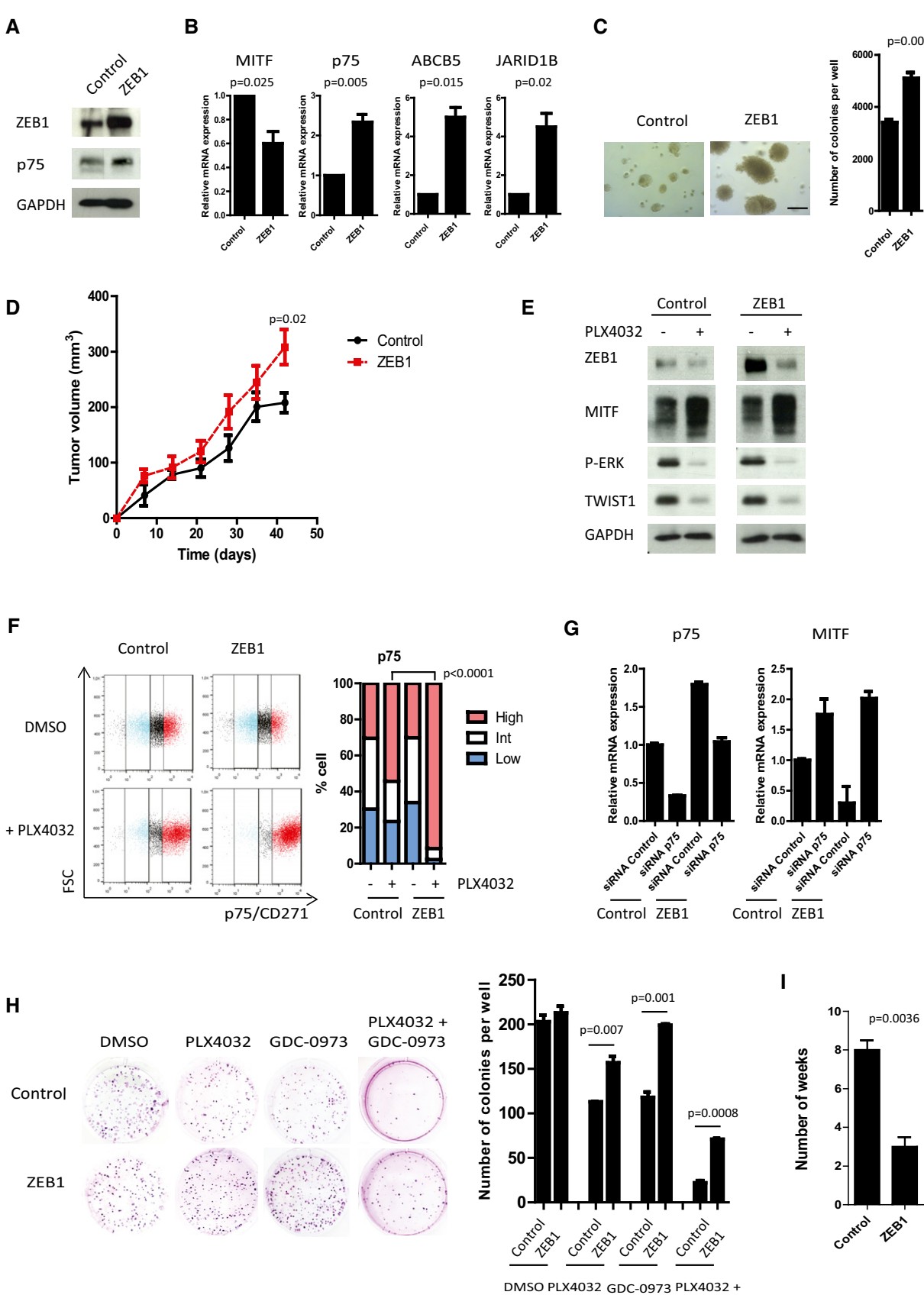

Figure 4.

◀

**Figure 4.  *ZEB1* overexpression in A375 melanoma cells potentiates the conversion into a MITF^low/p75^high stem-like tumor initiating phenotype, and promotes resistance to MAPKi.**

A   A375 cells were infected with retroviruses expressing *ZEB1*. Western blot analyses of ZEB1 and p75. GAPDH was used as a loading control.

B   Quantitative PCR analyses of *MITF*, *p75*, *ABCB5*, and *JARID1B* upon *ZEB1* expression. mRNA expression levels are represented relative to control cells, in which the levels were fixed at 1 (mean ± SD, *n* = 3, Student's *t*-test).

C   Soft agar colony formation assay upon *ZEB1* expression. Scale bar = 200 μm. Histograms represent quantitative analyses (mean ± SD, *n* = 3, Student's *t*-test).

D   2 × 10^6 control or *ZEB1*-overexpressing A375 cells were injected subcutaneously into nude mice. The mean tumor volume for five mice is represented ± SEM (Student's *t*-test, *P*-value at 42 days).

E   Western blot analyses of ZEB1, MITF, P-ERK, and TWIST1 levels in control or *ZEB1*-expressing cells ± 150 nM PLX4032 treatment for 24 h. GAPDH was used as a loading control.

F   FACS analyses of p75 cell surface expression upon *ZEB1* overexpression, after 10 days with or without 150 nM PLX4032 treatment. Bar chart representing the mean percentage of p75^high, p75^int, and p75^low cells from two independent experiments (Fisher's exact test).

G   Control or ZEB1-overexpressing A375 cells were transfected with control or p75-siRNA. *p75* and *MITF* expression levels were analyzed by quantitative PCR after 48 h. mRNA expression levels are represented relative to cells transfected with control siRNA, in which the levels were fixed at 1 (mean ± SD, *n* = 2).

H   Clonogenic assay ± PLX4032 (150 nM), ± GDC-0973 (5 nM) treatment for 10 days. The graphs represent the mean number of colonies (± SD) in three independent experiments (Student's *t*-test).

I   Number of weeks of chronic exposure to PLX4032 before emergence of resistance in control or ZEB1-expressing cells (mean ± SD, *n* = 3, Student's *t*-test).

Source data are available online for this figure.

(Fig 4G). The knockdown of *p75* in *ZEB1*-overexpressing A375 cells resulted in a level of *p75* equivalent to that in control cells. The knockdown of *p75* induced an increase in *MITF* expression levels in *ZEB1*-expressing cells similar to that in control cells, thus suggesting that ZEB1-mediated downregulation of *MITF* is dependent on p75. We conclude that p75 is at least responsible for part of the effects associated with ZEB1.

Next, we showed that the ability of *ZEB1*-overexpressing A375 and SKMEL5 cells to grow in a clonogenic assay was only moderately affected in the presence of PLX4032 over the 10-day experimental time course, whereas the number of colonies drastically decreased in the case of the control cells (Fig 4H, Appendix Fig S3E). Concordantly, chronic treatment of A375 and SKMEL5 cells with PLX4032 demonstrated that *ZEB1* overexpression favored the emergence of resistance *in vitro*. Indeed, in the presence of ZEB1, the PLX4032 dose had to be increased more rapidly and led to the emergence of resistant clones after only 3 weeks, in comparison with the 8 weeks required for the control cells (Fig 4I). Finally, since MEKi are now routinely used in clinical applications in combination with BRAFi, we investigated whether ZEB1 could also promote resistance to the combined PLX4032 and GDC-0973 (cobimetinib) treatment, and found that *ZEB1*-expressing A375 (Fig 4H) and SKMEL5 cells (Appendix Fig S3E) were also more resistant to this combined treatment. Collectively, these data indicated that the ectopic expression of *ZEB1* drives the emergence of resistance upon chronic exposure to MAPKi by exacerbating a MITF^low/p75^high stem-like phenotype insensitive to treatment.

We then investigated the consequences of ectopically expressing *ZEB1* in ZEB1^low/MITF^high cells. Surprisingly, in the ZEB1^low/MITF^high cell lines, such as 501MEL, *ZEB1* overexpression was not sufficient to promote p75 expression, even upon PLX4032 treatment (Fig EV2A–C). However, *ZEB1* expression increased the clonogenic growth of ZEB1^low 501MEL and, whereas PLX4032 treatment drastically inhibited the growth of control cells in soft agar, *ZEB1*-overexpressing cells were less sensitive to the BRAFi in this assay (Fig EV2D), suggesting that ZEB1 can promote resistance in this model without p75 induction.

Finally, to investigate the function of ZEB1 in physiological models with low levels of ZEB1 expression, the EMT inducer was ectopically expressed in two *BRAF*^V600 patient-derived short-term

cultures, C-09.10 cells and GLO (Figs 5 and EV3). *ZEB1* ectopic expression in C-09.10 cells led to a significant decrease in *MITF* levels and increase in *p75* levels (Fig 5A, B and D). Similarly in GLO cells, *ZEB1* ectopic expression was shown to promote the conversion into a p75^high state, which was potentiated upon PLX4032 treatment (Fig EV3). *ZEB1*-induced phenotype switching was associated with an increased capacity to form colonies in soft agar (Fig 5C) and to resistance to BRAFi as assessed by a clonogenic assay in the presence of PLX4032 (Fig 5E). Results in these two short-term culture models therefore validated the conclusions obtained in established ZEB1^high cell lines, in a ZEB1^low context.

### ZEB1 knockdown in BRAF^V600 melanoma cells promotes cell differentiation and inhibits tumor growth

To assess the benefit of targeting ZEB1 as a therapeutic strategy, *ZEB1* was knocked down in the ZEB1^high A375 and SKMEL5 cell lines by infecting cells with an shRNA-*ZEB1*-encoding retrovirus. Proliferation of the cells was not affected, and neither cellular senescence nor apoptosis was observed. An increase in *MITF* as well as *ZEB2* and E-cadherin expression levels, and a decrease in *p75*, *ABCB5*, and *JARID1B* levels were observed upon *ZEB1* knockdown in A375 (Fig 6A–C and Appendix Fig S6) and SKMEL5 cells (Fig EV4). *ZEB1* knockdown triggered the conversion into a p75^low profile, as assessed by flow cytometry (Figs 6C and EV4B). Moreover, a fivefold decrease in the number of colonies growing in soft agar was observed upon *ZEB1* knockdown in those cell lines (Figs 6D and EV4C). We then examined whether the presence of ZEB1 was a requirement for the growth of malignant melanoma cells *in vivo*. As expected from previous data in melan-a and B16F10 murine cells (Caramel *et al*, 2013; Dou *et al*, 2014), *ZEB1* knockdown in A375 cells prevented tumor initiation in nude mice (Fig 6E), clearly demonstrating that ZEB1 is required for the tumorigenic capacity of melanoma cells. We thus used IPTG-inducible shRNA-*ZEB1* to evaluate the impact of *ZEB1* knockdown on tumor shrinkage in established tumors. Two *ZEB1*-targeted shRNAs that consistently reduced the levels of endogenous ZEB1 protein upon IPTG treatment *in vitro* were used (Fig 6F). The A375 cells infected with the IPTG-inducible shRNA control or shRNA-*ZEB1* were injected subcutaneously into nude mice. When tumors reached a

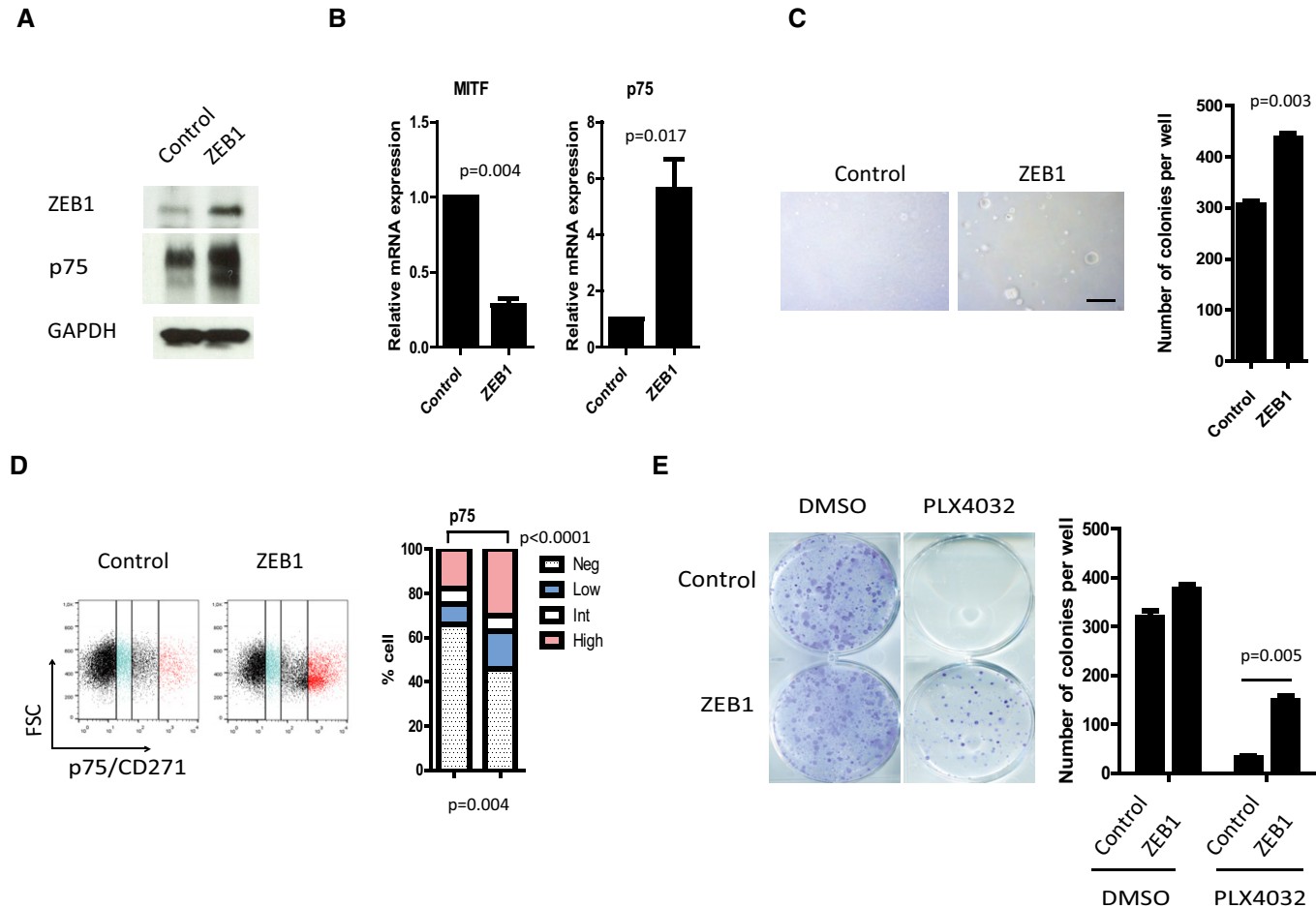

**Figure 5. ZEB1 overexpression in patient-derived ZEB1$^{low}$/MITF$^{high}$ short-term culture cells promotes the conversion into a MITF$^{low}$/p75$^{high}$ stem-like phenotype, resistant to MAPKi.**

C-09.10 short-term culture cells were infected with retroviruses expressing *ZEB1*.

A Western blot analyses of ZEB1 and p75. GAPDH was used as a loading control.
B Quantitative PCR analyses of *MITF* and *p75* upon ZEB1 expression. mRNA expression levels are represented relative to control cells (mean ± SD, n = 3, Student's *t*-test).
C Soft agar colony formation assay following *ZEB1* expression. Scale bar = 200 μm. Histograms represent quantitative analyses (mean ± SD, n = 3, Student's *t*-test).
D FACS analyses of p75 cell surface expression upon *ZEB1* overexpression. Bar chart representing the mean percentage of p75$^{high}$, p75$^{int}$, p75$^{low}$, and p75$^{negative}$ cells from two independent experiments (Fisher's exact test).
E Clonogenic assay ± PLX4032 (100 nM) treatment for 10 days. The graph represents the mean number of colonies (± SD) in three independent experiments (Student's *t*-test).

Source data are available online for this figure.

diameter of 5 mm, the mice were given IPTG in their drinking water. *ZEB1* knockdown led to a significant decrease in tumor growth, confirming the potent anti-tumor effect of ZEB1 inhibition (Fig 6F). Finally, to demonstrate the reversibility of the ZEB1-mediated phenotype switching, the expression of shRNA-*ZEB1* was induced for 10 days (+IPTG) and ZEB1 expression was then reversed by removing IPTG for 10 days (−IPTG). Upon IPTG withdrawal, levels of ZEB1, MITF, and p75 returned to the basal levels in untreated cells (Fig 6G). Taken together with the results obtained following *ZEB1* overexpression, our data indicate that ZEB1 drives the reversible conversion of MITF$^{high}$/p75$^{low}$ differentiated into MITF$^{low}$/p75$^{high}$ stem-like/initiating phenotypes, and regulates the subsequent tumorigenic capacity of melanoma cells.

**ZEB1 knockdown sensitizes naive melanoma cells to BRAFi and decreases the viability of BRAFi-resistant melanoma cells**

Next, we investigated whether knocking down *ZEB1* in initially PLX4032-naive melanoma cell lines could increase their sensitivity to this drug. *ZEB1* knockdown in sensitive ZEB1$^{high}$ A375 and SKMEL5 melanoma cells inhibited colony formation in soft agar, to a similar extent to that observed with BRAF inhibition (60%). Importantly, the colony number was reduced by 90% in cells knocked down for *ZEB1* following PLX4032 treatment, indicating a synergistic effect upon combined inhibition of BRAF and ZEB1 (Figs 7B and EV4C). This was associated with a concomitant decrease in the size of the colonies. We then analyzed the

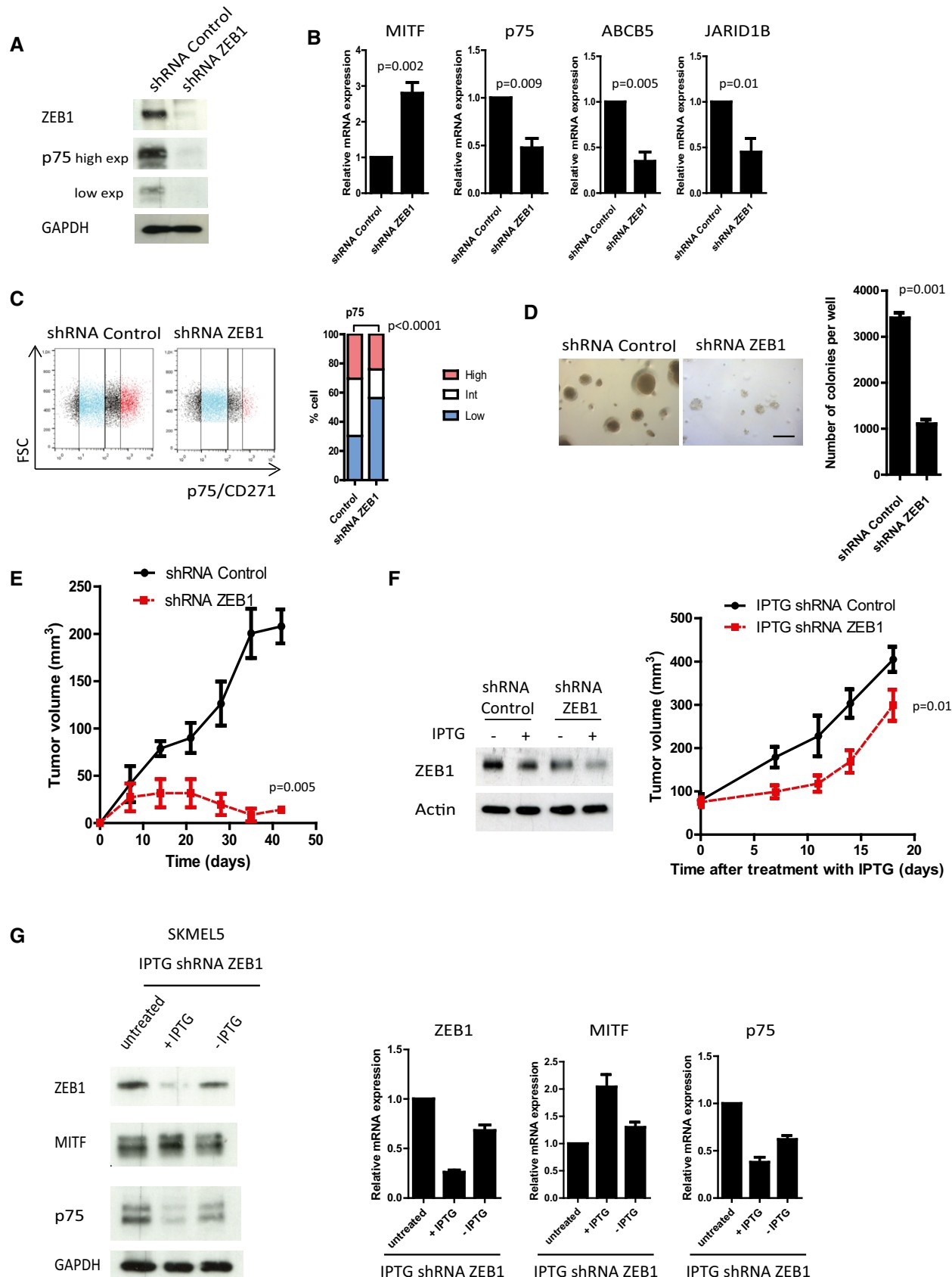

**Figure 6.**

◀

**Figure 6.  *ZEB1* knockdown in ZEB1<sup>high</sup>/MITF<sup>low</sup> melanoma cells promotes the reversible conversion into a MITF<sup>high</sup>/p75<sup>low</sup> differentiated phenotype and inhibits tumor growth *in vivo*.**

A375 cells were infected with retroviruses expressing a shRNA control or shRNA-*ZEB1*.

A   Western blot analyses of ZEB1 and p75 upon *ZEB1* knockdown. GAPDH was used as a loading control. High and low exposures (exp) for p75 are shown.

B   Quantitative PCR analyses of *MITF*, *p75*, *ABCB5*, and *JARID1B* upon *ZEB1* knockdown. mRNA expression levels are represented relative to shRNA control cells (mean ± SD, *n* = 3, Student's *t*-test).

C   FACS analyses of p75 expression upon *ZEB1* knockdown. Bar chart representing the mean percentage of p75<sup>high</sup>, p75<sup>int</sup>, and p75<sup>low</sup> cells from two independent experiments (Fisher's exact test).

D   Soft agar colony formation assay upon *ZEB1* knockdown. Scale bar = 200 μm. Histograms represent quantitative analyses (mean ± SD, *n* = 3, Student's *t*-test).

E   2 × 10$^6$ shRNA control or shRNA-*ZEB1* A375 cells were injected subcutaneously into nude mice. The mean tumor volume for five mice is represented (± SEM) (Student's *t*-test, *P*-value at 42 days).

F   A375 cells were infected with an IPTG-inducible shRNA-*ZEB1*. Left panel: Western blot analyses of ZEB1 expression ± IPTG (100 μM) treatment for 6 days. Actin was used as a loading control. Right panel: 2 × 10$^6$ IPTG-inducible shRNA control or shRNA-*ZEB1* A375 cells were injected subcutaneously into nude mice. When the tumor reached 5 mm in diameter, ZEB1 expression was silenced by adding IPTG (10 mM) into the drinking water for 20 days. The mean tumor volume for five mice is represented (± SEM). (Student's *t*-test, *P*-value at 18 days).

G   SKMEL5 cells expressing an IPTG-inducible shRNA-*ZEB1* were treated with IPTG (100 μM) for 10 days (+IPTG), and then IPTG was removed (-IPTG) and ZEB1, MITF, and p75 expression levels were analyzed by Western blot and/or quantitative PCR analyses. GAPDH was used as a protein loading control, and mRNA expression levels are represented relative to untreated cells, in which the levels were fixed at 1 (mean ± SD, *n* = 2).

Source data are available online for this figure.

efficacy of such a combination *in vivo* in SKMEL5 xenografts. Mice were treated with vemurafenib through daily oral administration. *ZEB1* knockdown alone was as efficient as vemurafenib treatment in decreasing SKMEL5 xenografted tumor growth, but the combined inhibition of BRAF and ZEB1 did not lead to a further significant decrease in tumor growth (Fig EV4D). This finding does not preclude the hypothesis that ZEB1 inhibition prevents the emergence of resistant cells *in vivo*. Indeed, we could show *in vitro* that *ZEB1* knockdown prevented the emergence of resistance upon chronic exposure to PLX4032 in A375 cells, with a significant delay until the emergence of resistant cells (Fig 7C).

Finally, we investigated whether ZEB1 expression was necessary for the survival of resistant melanoma cells. The consequences of *ZEB1* knockdown were analyzed in a cell line intrinsically resistant to vemurafenib, the RPMI7951, which exhibits high levels of ZEB1 expression (Fig 1, Appendix Fig S1) as well as in the vemurafenib-acquired resistant cell lines A375-R and SKMEL5-R, and patient-derived GOKA and ESP cells. ZEB1 expression was rapidly downregulated following PLX4032 treatment in the parental naive A375 cells (Fig 7A), whereas a high level of ZEB1 expression was maintained in the resistant A375-R cells in the presence of 3 μM PLX4032 (Fig 7D). In all these models, *ZEB1* knockdown increased the sensitivity to PLX4032, resulting in a decreased clonogenic capacity (Fig 7E), which was associated with an apoptotic response upon co-treatment with BRAFi, as shown by an increase in the abundance of cleaved PARP in A375-R (Fig 7D). Moreover, p75 expression was decreased and *MITF* expression was increased upon *ZEB1* knockdown in resistant ESP cells (Fig 7F). shRNA control or shRNA-*ZEB1* ESP cells were then xenografted into nude mice and orally treated with IPTG ± vemurafenib. While vemurafenib did not affect tumor growth of control resistant cells, ZEB1 inhibition alone or in combination with vemurafenib led to a significant decrease in tumor growth (Fig 7G). The efficacy of *ZEB1* knockdown *in vivo* was assessed by Western blot and immunostaining directly in the tumors (Fig 7H). Overall, these data demonstrate that *ZEB1* knockdown decreases the viability of resistant melanoma cells in both MITF<sup>low</sup> (RPMI7951, A375-R, GOKA) and MITF<sup>high</sup> (SKMEL5-R, ESP) cellular contexts.

## Discussion

The aberrant activation of an epithelial–mesenchymal transition and the subsequent generation of a cancer stem cell phenotype are generally believed to foster therapy resistance in carcinoma (Singh & Settleman, 2010). Such a relationship was not previously investigated in melanoma, where the characterization of resistance mechanisms to BRAF and/or MEK inhibitors remains a major issue. Melanomas harbor a high level of intra-tumoral heterogeneity that relies on exacerbated cell plasticity, which drives the highly efficient reversible conversion between non-tumorigenic and tumorigenic states (Quintana *et al*, 2008, 2010; Meacham & Morrison, 2013). According to the concept of phenotype switching, tumor progression does not necessarily rely on clonal evolution but rather on the reversible reprogramming of signaling networks in large populations of cells. In parallel to "genetic" resistance to MAPKi, due to the acquisition of genetic mutations, affecting *NRAS* or *MEK* (Nazarian *et al*, 2010; Van Allen *et al*, 2014), the present study emphasizes melanoma cell plasticity as a potent driver of "phenotypic" resistance (Roesch, 2015). Our findings demonstrate that the ZEB1 transcription factor is a key determinant of melanoma phenotypic plasticity, tumorigenicity, and resistance to MAPKi, by fostering the adaptation to the therapeutic drugs. Data from cell lines and patients indicate that a subset of mutated BRAF melanomas with high levels of ZEB1 expression may be intrinsically insensitive to BRAFi and MEKi, suggesting that melanoma patients with high levels of ZEB1 expression may not benefit from MAPKi treatment. Analyses of larger patient cohorts are required to validate the use of ZEB1 as a predictive marker in order to stratify *BRAF*-mutated melanoma into MAPKi-sensitive and MAPKi-resistant subgroups. Moreover, ZEB1 increased expression is frequently observed after acquisition of resistance following chronic treatment with BRAFi. Functional studies revealed that *ZEB1* overexpression is sufficient to drive the emergence of resistance to BRAFi alone or in combination with MEKi, whereas *ZEB1* inhibition sensitizes naive melanoma cells to BRAFi, prevents the emergence of resistance upon chronic exposure *in vitro*, and decreases the viability of resistant cells. Since IC$_{50}$ values for PLX4032 were only moderately modified upon *ZEB1* over-expression after 3 days of treatment in sensitive A375, C-09.10, or

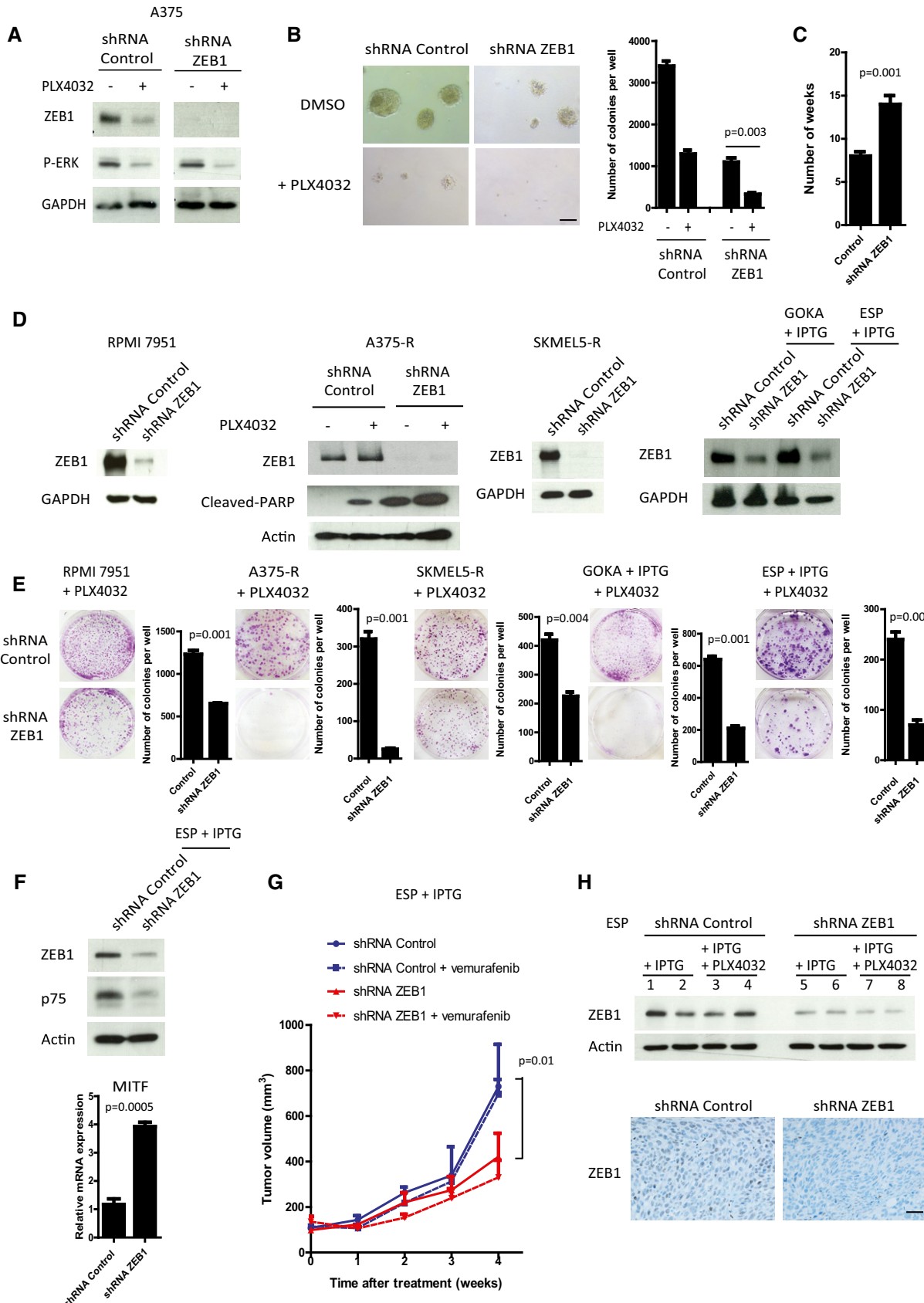

**Figure 7.**

◀ **Figure 7.  ZEB1 knockdown sensitizes naive melanoma cells to BRAFi and induces cell death in BRAFi-resistant melanoma cells.**

A   Western blot analyses of ZEB1 and P-ERK in shRNA control or shRNA-*ZEB1*-expressing A375 cells ± 150 nM PLX4032 treatment for 24 h. GAPDH was used as a loading control.

B   Soft agar colony formation assay in A375 cells in the presence or absence of PLX4032 (150 nM). Scale bar = 100 μm. Histograms represent quantitative analyses (mean ± SD, *n* = 3, Student's *t*-test).

C   Number of weeks of chronic exposure to PLX4032 before emergence of resistance in shRNA control or shRNA-*ZEB1*-expressing cells (mean ± SD, *n* = 3, Student's *t*-test).

D   RPMI7951, A375-R, and SKMEL5-R were infected with a retrovirus encoding a constitutive shRNA-*ZEB1*. Short-term cultures of GOKA and ESP cells, derived from vemurafenib-resistant patients, were infected with a lentivirus encoding an IPTG-inducible shRNA-*ZEB1*. Western blot analyses showing efficient *ZEB1* knockdown in the different models, ± IPTG (200 μM), ± PLX4032 (3 μM) as indicated. Induction of cell death was assessed by PARP cleavage. GAPDH or actin was used as loading control.

E   Clonogenic assays in the presence of 3 μM PLX4032, and with or without IPTG (200 μM) as indicated. The graphs represent the mean number of colonies (± SD) in three independent experiments (Student's *t*-test).

F   Western blot analyses of ZEB1 and p75 and quantitative PCR analyses of *MITF* in shRNA control or shRNA-*ZEB1* ESP vemurafenib-resistant patient-derived short-term culture cells. Actin was used as a loading control. mRNA expression levels are represented relatively to control cells (mean ± SD, *n* = 3, Student's *t*-test).

G   2.5 × 10⁶ shRNA control or shRNA-*ZEB1* vemurafenib-resistant ESP cells were injected subcutaneously in nude mice. When the tumor reached 5 mm in diameter, ZEB1 expression was silenced by providing mice with IPTG (10 mM) in their drinking water and orally administering vemurafenib (50 mg/kg) daily for 4 weeks. The mean tumor volume for five mice is represented (± SEM) (Student's *t*-test, *P* at 4 weeks).

H   Upper part: Western blot analyses of ZEB1 in shRNA control (1 to 4) or shRNA-*ZEB1* (5 to 8) ESP xenograft tumors, showing efficient *ZEB1* knockdown directly in the tumors, after IPTG ± PLX4032 treatment *in vivo*. Lower part: Representative pictures of ZEB1 immunostaining in shRNA control or shRNA-*ZEB1* tumors, after IPTG treatment *in vivo*. Scale bar = 40 μm.

Source data are available online for this figure.

GLO cells (Appendix Fig S7), this further indicates that ZEB1 effects rely on a process of drug-induced phenotype adaptation that requires at least one week of treatment. Moreover, cell death was observed upon *ZEB1* knockdown in A375-R cells even in the absence of PLX4032 treatment (Fig 7D), indicating that these resistant cells are addicted to ZEB1.

Our *in vitro* investigation of the underlying mechanisms uncovered that ZEB1 regulates the reversible transitions from a drug-sensitive differentiated state to a drug-resistant stem-like phenotype, associated with a downregulation of MITF and direct or indirect activation of several melanoma-initiating cell markers, including the neural crest cell marker p75/CD271, a crucial determinant of the colony formation of melanoma cells, and of tumorigenicity (Boiko *et al*, 2010; Civenni *et al*, 2011; Redmer *et al*, 2014). However, knockdown experiments demonstrated that p75 was only responsible for part of the effects associated with ZEB1, since the ZEB1-mediated decrease in *Tyrosinase* was not reverted after the knockdown of *p75*. Moreover, *ZEB1* expression promoted resistance to BRAFi in 501MEL cells without induction of p75, suggesting that ZEB1 can promote resistance by other mechanisms. The expression of two other markers that do not necessarily overlap with p75 (Cheli *et al*, 2014), namely the ABCB5 transporter and the histone demethylase JARID1B (Schatton *et al*, 2008; Roesch *et al*, 2010), was also regulated upon ZEB1 modulation. Overall, ZEB1, as a transcription factor which can act as a transcriptional repressor or activator thanks to the binding to specific cofactors, is responsible for the modulation of a large panel of targets, including downregulation of melanocyte differentiation markers and upregulation of melanoma-initiating cell markers that cooperate in mediating resistance to MAPKi.

*ZEB1* ectopic expression was sufficient to increase p75 levels in ZEB1^high established cell lines (A375, SKMEL5), and ZEB1^low patient-derived short-term cultures (C-09.10 and GLO), and this effect was potentiated upon BRAFi treatment. However, in ZEB1^low/MITF^high established melanoma cell lines, such as 501MEL, *ZEB1* overexpression was not sufficient to promote p75 expression, even upon PLX4032 treatment. This may be due to epigenetic modifications acquired during the establishment of these cell lines that may block the promoter in a closed chromatin configuration.

We found a potent correlation between high levels of ZEB1 and low levels of MITF expression in various settings in both cell lines and patient-derived samples. The ZEB1^high/MITF^low profile was associated with intrinsic resistance to MAPKi. Invasive MITF^low cells with high levels of expression of WNT5A or of the receptor tyrosine kinase AXL were recently shown to be more resistant to MAPKi (Anastas *et al*, 2014; Konieczkowski *et al*, 2014; Muller *et al*, 2014), supporting our own model. AXL, a known target of FRA1 that promotes EMT in epithelial cells (Sayan *et al*, 2012), exhibit high levels of expression in our resistant melanoma models (Appendix Fig S8). However, neither AXL nor WNT5A levels were induced upon *ZEB1* ectopic expression, suggesting that ZEB1 function in resistance may be independent of these pathways. ZEB1 is obviously not the only factor influencing sensitivity to MAPKi treatment and we do not exclude the putative co-occurrence of other mechanisms of resistance that could account for the differential sensitivity to treatment observed in established cell lines *in vitro*. Both ZEB1 and MITF are at least in part regulated by the MAPK pathway. Although ZEB1 is downregulated by PLX4032 at an early stage, it is strongly upregulated in resistant cells, indicating drug adaptation. We previously showed that ZEB1 expression levels are not only regulated at the transcriptional level by the FRA1 transcription factor, but also at the posttranslational level (Caramel *et al*, 2013). This may explain why ZEB1 levels are still decreasing in cells that ectopically express ZEB1 upon PLX4032 treatment. MITF is upregulated after a short treatment with PLX4032 but may be lost in resistant cells, in favor of a role for lower MITF expression in resistance to MAPKi. However, the role of MITF is complex, since both high and low levels of MITF can be found in cells with acquired resistance, even within the same tumor in different clones, suggesting that MITF may or may not be required for the acquisition and maintenance of resistance to MAPKi (Wellbrock & Arozarena, 2015). Concordantly, we observed that cell lines and patient-derived samples with intrinsic or acquired resistance to MAPKi could display a ZEB1^high/MITF^high profile. Furthermore, *ZEB1* knockdown decreased the viability of resistant melanoma cells in both MITF^low and MITF^high contexts, suggesting that ZEB1 partly functions through MITF-independent mechanisms.

High levels of ZEB1 expression undoubtedly promote an invasive phenotype in melanoma, including decreased *E-cadherin* and increased expression of *Vimentin, SPARC,* and *MMPs* (Caramel *et al*, 2013; Verfaillie *et al*, 2015) (Appendix Fig S4). Our results reinforce the notion that ZEB1 additionally displays intrinsic oncogenic functions. ZEB1 is expressed in the bulk of primary melanoma. Its ectopic expression promotes stemness and tumorigenic features in melanoma cell lines, and its knockdown drastically decreases the tumorigenic growth of melanoma cells *in vivo*. Our results thus demonstrate that ZEB1 is a major driver of phenotype switching-mediated resistance to MAPKi and highlight that it is not the EMT phenotype or invasive status itself but the specific functions of EMT-TFs that cause resistance in melanoma cells. In contrast, ZEB2 is associated with low levels of ZEB1 expression and a differentiated MAPKi-sensitive phenotype. As previously reported for their respective tumorigenic capacities (Caramel *et al*, 2013), ZEB2 could therefore play an antagonistic function to ZEB1 in terms of resistance to treatment in melanoma. Interestingly, specific functions of EMT-TFs in resistance to treatment were previously reported in carcinoma, corroborating our own model. As an example, ZEB1 was previously shown to induce radioresistance through EMT-independent, Chk1-dependent mechanisms (Zhang *et al*, 2014, 2015). Moreover, invasive phenotypes and stemness properties can be uncoupled in carcinomas, since ZEB1 and TWIST1 exhibit a dose-dependent role in malignant progression. A lower threshold of ZEB1 or TWIST1 is sufficient to induce stemness and tumor initiation, whereas further induction is necessary for EMT induction, invasion, and tumor metastasis (Liu *et al*, 2014; Beck *et al*, 2015). Of note, while the ectopic expression of *ZEB1/ZEB2* has been associated with attenuated cell proliferation in carcinoma models (Mejlvang *et al*, 2007), we did not observe any proliferation defect upon *ZEB1* expression in melanoma cells, further highlighting cell-type-specific functions of ZEB1 and ZEB2 in melanoma compared to carcinoma, as we previously reported (Caramel *et al*, 2013; Puisieux *et al*, 2014).

While our study was mainly focused on ZEB1, TWIST1 is also frequently activated in melanoma. However, the ectopic expression of *TWIST1* did not significantly stimulate the colony formation capacity of melanoma cell lines (Appendix Fig S9), suggesting that ZEB1 is a stronger oncogenic factor than TWIST1 in this tumor type. Nevertheless, if the ectopic expression of *TWIST1* is unable to confer resistance to PLX4032 in ZEB1[high] cell lines (A375/SKMEL5), our results indicate that TWIST1 can drive resistance in a ZEB1[low] context (501MEL cells, Fig EV2). When combined with the observation that a significant number of primary resistant ZEB1-negative melanomas exhibit high levels of TWIST1 expression, these results suggest that ZEB1 is the main driver of BRAFi resistance, but that TWIST1 may complement ZEB1 when this factor is not activated, and may thus constitute a potential therapeutic target in a subset of melanomas.

Finally, a better understanding of the function of EMT-TFs in melanoma cell plasticity should lead to the design of novel combination therapies targeting specific EMT-TFs and the MAPK pathway. Even if the aberrant expression of ZEB1 is more frequently found in *BRAF*[V600]-mutated tumors (54%), it may be beneficial to target ZEB1 as a mediator of cell plasticity in a subset of *NRAS*[Q61R]-mutated (29%) or *BRAF* and *NRAS* WT tumors (16%) with high levels of ZEB1 expression (Appendix Fig S2). ZEB1 inhibition was

shown to increase sensitivity to MAPKi, prevent the emergence of resistance, and induce cell death in various melanoma cells with intrinsic or acquired resistance to BRAFi. Since the targeting of transcription factors is challenging, a strategy may consist in identifying ZEB1 gene targets or epigenetic mechanisms regulating ZEB1-driven plasticity to overcome ZEB1-associated drug resistance. In this context, the class I HDAC inhibitor mocetinostat was recently shown to inhibit the expression of ZEB1, to upregulate its target, miR-203, and to restore sensitivity to chemotherapy in pancreatic cancer cells (Meidhof *et al*, 2015).

# Materials and Methods

## Human tumor samples and immunohistochemical analyses

Melanoma tumor samples were obtained through the Biological Resource Center of the Hôpital Lyon Sud, Hospices Civils de Lyon, and of the Hôpital Saint Louis, AP-HP, Paris. Human tumor samples were used with the patient's written informed consent. This study was approved by the scientific board of the Hospices Civils de Lyon. Immunohistochemical studies were conducted on a cohort of 70 melanomas with a *BRAF*[V600] mutation, for which clinical data and response to vemurafenib ± cobimetinib treatment were known. Of the 70 patients, 16 received the combination therapy. Patients were referred to as "initially responding" when regression of the tumor was observed (40) and referred to as "initially non-responding" when the tumor progressed or remained stable (30) according to RECIST criteria. In addition, biopsies were performed before and after vemurafenib treatment ($n = 12$), on non-responding ($n = 3$) or relapse ($n = 9$) patients. Specimens were formalin-fixed, paraffin-embedded, and 3-μm-thick tissue sections were cut. The sections underwent IHC staining using steam heat-induced epitope retrieval, the Ventana Benchmark XT platform (Ventana-Roche Tissue Diagnostics, Meylan France), Ultraview red detection and commercially available antibodies against ZEB1 (H102, rabbit polyclonal, 1/800, Santa Cruz), MITF (C5/D5, mouse monoclonal, 1/200, Roche), and TWIST1 (Twist2C1a, mouse monoclonal, 1/50, Abcam, Cambridge, MA, USA). For all three antibodies, the staining was nuclear. A blinded evaluation of the staining was carried out by experienced pathologists. The IHC staining was scored evaluating the intensity (1, 2, 3) and the percentage of positive cells.

## CCLE and TCGA data sets

### Data set analyses

The levels of mRNA expression of 61 melanoma cancer cell lines obtained from the "*CCLE_Expression_Entrez_2012-10-18.res*" file available through the Cancer Cell Line Encyclopedia (CCLE) were analyzed. In addition, the drug response profile of 28 *BRAF*[V600]-mutated melanoma cancer cell lines obtained from the "*CCLE_NP24.2009_Drug_data_2015.02.24.csv*" file (CCLE) was also analyzed (Barretina *et al*, 2012).

Furthermore, the levels of mRNA expression of 467 human melanomas obtained from the level 3 data *UNC IlluminaHiSeq_RNASeqV2* of the Skin Cutaneous Melanoma study provided by The Cancer Genome Atlas (TCGA) were analyzed. The mutational status of *BRAF* and *NRAS* genes of 341 human

melanomas obtained from the cBioPortal (http://www.cbioportal.org/) were also analyzed (Cerami *et al*, 2012; Gao *et al*, 2013; The Cancer Genome Atlas Network, 2015). TCGA level 3 RNA sequencing data were normalized using the DESeq method with the R software (version 3.1.2; the R core Team R: a language and environment for statistical computing, 2008; http://www.R-project.org) (Anders & Huber, 2010).

### TCGA mRNA expression data processing

For the TCGA samples, the levels of *ZEB1* mRNA expression exceeding the 80[th] percentile were referred to as "High", while *ZEB1* expression levels inferior to the 20[th] percentile were referred to as "Low". The remaining samples were classified as "Intermediate".

### Mouse injections

Experiments using mice were performed in accordance with the animal care guidelines of the European Union and French laws and were validated by the local Animal Ethic Evaluation Committee (CECCAPP). Mice were housed and bred in a specific pathogen free animal facility "AniCan" at the CRCL, Lyon, France. Single cell suspensions of A375 cell lines ($2 \times 10^6$ cells), SKMEL5 ($2 \times 10^6$ cells), or ESP ($2.5 \times 10^6$ cells) in PBS/Matrigel (BD Biosciences, Oxford, UK) (1/1) were injected subcutaneously into the flank of six-week-old female athymic Swiss nude mice (Charles River Laboratories). Five mice were included in each experimental group, in separate cages (IPTG alone or IPTG + vemurafenib). No blinding was done. For the IPTG-inducible models, when the tumor reached 5 mm in diameter, ZEB1 expression was silenced by adding IPTG (10 mM) (Sigma) to the drinking water. BRAF was inhibited by orally administering vemurafenib (50 mg/kg) or vehicle (DMSO/PBS) daily for 2–5 weeks. Tumor growth was monitored during 4–6 weeks post-injection. Tumors grew up to 1.5 cm in diameter, at which point animals were euthanized. Tumors were embedded in paraffin and ZEB1 staining was performed using the anti-ZEB1 antibody (IHC-00419, 1/300, Bethyl), and DAB detection and counterstaining with hematoxylin.

### Cell culture and reagents

WM115, RPMI7951, SKMEL24, WM793, SKMEL28, SKMEL5, A375, SKMEL3, and MALM3M human melanoma cell lines were purchased from ATCC. 501MEL and WM9 cell lines were kindly provided by Dr Robert Ballotti and M4BE cell line by Dr Thibault Voeltzel (Centre Léon Bérard). All these *BRAF^V600* human melanoma cell lines were cultured in DMEM complemented with 10% FBS (Cambrex) and 100 U/ml penicillin–streptomycin (Invitrogen). In order to authenticate the cell lines, the expected major genetic alterations were verified by NGS sequencing. The absence of mycoplasma contamination was checked every 3 weeks with the MycoAlert detection kit (Lonza). Patient-derived short-term cultures (< 10) were established from *BRAF^V600* metastatic melanomas, before treatment for GLO and C-09.10, or after acquisition of resistance to vemurafenib for ESP and GOKA. C-09.10 were kindly provided by Dr Robert Ballotti (Nice). These short-term cell cultures were grown in RPMI complemented with 10% FBS and 100 U/ml penicillin–streptomycin. PLX4032/vemurafenib and GDC0973/cobimetinib were purchased from Selleck Chemicals (Houston, TX,

USA) and reconstituted in DMSO. Generation of A375-R and SKMEL5-R resistant models was performed by treating cells chronically with increasing doses of PLX4032 for 2–3 months. All BRAFi-resistant cell lines were permanently cultured in the presence of 3 μM PLX4032.

### Retroviral and lentiviral infection

Human embryonic kidney 293T cells ($4 \times 10^6$) were transfected with retroviral or lentiviral expression constructs (10 μg) in combination with GAG-POL (5 μg) and ENV expression vectors (10 μg) using GeneJuice (Millipore). Viral stocks were collected 48 h post-transfection, filtered (0.45 μm membrane), and placed in contact with $2 \times 10^6$ melanoma cells for 8 h in the presence of 8 μg/ml polybrene. Forty-eight hours post-infection, cells were selected in the presence of puromycin (1 μg/ml) (Invitrogen). The following constructs: HA-ZEB1, Flag-TWIST1, shRNA control, and shRNA-*ZEB1* were described previously in pBABE-Puro vector (Caramel *et al*, 2013). IPTG-inducible shRNA control and shRNA-*ZEB1* (TRCN0000369266 and TRCN0000369267) in pLKO_IPTG_3xLacO were purchased from Sigma-Aldrich (St-Louis, MO, USA). For activation of shRNA expression, IPTG (Sigma) was added to the culture medium (100–200 μM) every two days. For reversion experiments, IPTG was removed and analyses performed after 10 days.

### siRNA transfections

For inactivation of *p75* by small interference RNA, $3 \times 10^5$ cells were seeded in 6-well plates. Control or p75-siRNA (Life Technologies) (Silencer select human NGFR s194654, #4392420) were transfected into A375 cells (with a final concentration of 50 pM) using Lipofectamine following the manufacturer's instructions (Life Technologies). mRNA were extracted 48 h after transfection.

### Immunoblot analyses

Cells were washed twice with phosphate-buffered saline (PBS) containing $CaCl_2$ and then lysed in a 100 mM NaCl, 1% NP40, 0.1% SDS, 50 mM Tris pH 8.0 RIPA buffer supplemented with a complete protease inhibitor cocktail (Roche, Mannheim, Germany) and phosphatase inhibitors (Sigma-Aldrich). Protein expression was examined by Western blot using the anti-ZEB1 (H102, 1/200, Santa Cruz), anti-ZEB2 (1/500, Sigma), anti-TWIST1 (Twist2C1a, 1/100, Abcam, Cambridge, MA, USA), anti-P-ERK1/2 (#9106, 1/2,000, Cell Signaling Technology, Danvers, MA, USA), anti-MITF (clone C5, ab80651, 1/500, Abcam), anti-FRA1 (sc-605, 1/400, Santa Cruz), anti-p75 (1/200, Alomone Labs, Jerusalem), anti-AXL (AF154, 1/1,000, R&D Systems), and anti-PARP (cleaved form, 29 kDa) (ab6079, 1/200, Abcam) antibodies for primary detection. Loading was controlled using the anti-β-actin (clone AC-15, 1/10,000, Sigma-Aldrich), anti-GAPDH (1/20,000, Millipore), or anti-α-tubulin (T5168, 1/5,000, Sigma-Aldrich) antibodies. Horseradish peroxidase-conjugated rabbit anti-mouse, goat anti-rabbit, and donkey anti-goat polyclonal antibodies (Dako, Glostrup, Denmark) were used as secondary antibodies. Western blot detections were conducted using the Luminol reagent (Santa Cruz). For ZEB1 level analyses in mouse tumors, proteins were

extracted from frozen tumors in liquid nitrogen by homogenizing tissue in RIPA buffer.

## Q-PCR

Total RNA was isolated using RNeasy Kit (QIAGEN) and reverse-transcribed using a high cDNA capacity reverse transcription kit following the manufacturer's instructions (Fisher Scientific). Real-time PCR intron-spanning assays were designed using the ProbeFinder software (Roche). All reactions, including no-template controls and RT controls, were performed in triplicate on a CFX96 (Bio-Rad) and were analyzed with the Bio-Rad CFX manager software. Human *HPRT1* was used for normalization. The primers used were as follows: human *ZEB1* AGG GCA CAC CAG AAG CCA G and GAG GTA AAG CGT TTA TAG CCT CTA TCA; human *ZEB2* AAG CCA GGG ACA GAT CAG C and GCC ACA CTC TGT GCA TTT GA; human *TWIST1* GGC TCA GCT ACG CCT TCT C and CCT TCT CTG GAA ACA ATG ACA TCT; human *MITF* CAT TGT TAT GCT GGA AAT GCT AGA and TGC TAA AGT GGT AGA AAG GTA CTG C; human *p75*, TCA TCC CTG TCT ATT GCT CCA and TGT TCT GCT TGC AGC TGT TC; human *ABCB5* TTG AAA CCT TCG CAA TAG CC and TGG AAA AGT TAT CTA TAC TGG GTT TCT; human *JARID1B* AGC AGA CTG GCA TCT GTA AGG and GAA GTT TAT CAA CAT CAC ATG CAA; and *HPRT1* TGA CCT TGA TTT ATT TTG CAT ACC and CGA GCA AGA CGT TCA GTC.

## Flow cytometry analyses

To analyze the expression of the p75/CD271 cell surface markers, cells were incubated with an Alexa Fluor 647-conjugated anti-CD271 antibody (BD Pharmingen) for 1 h before being counted on a FACSCalibur. Data were analyzed using the FlowJo 7.5.5 software.

## Soft agar colony formation assay

Melanoma cell lines were transduced with cDNA or shRNA retroviral or lentiviral expression vectors and selected with puromycin. Plates were prepared by coating with 0.75% low-melting agarose (Lonza) in growth medium and then overlaid with cell suspension in 0.45% low-melting agarose ($5 \times 10^3$ cells/well). Plates were incubated for 2–3 weeks at 37°C. Colonies were stained with crystal violet (1 mg/ml; Sigma-Aldrich) and counted under microscope.

## Viability assays

For short-term viability assays, the CellTiter-Glo Luminescent Cell Viability Assay (ATP assay) (Promega) was used, based on quantitation of the ATP present, which signals the presence of metabolically active cells. 1,000 cells in 96-well plates were treated with the indicated drugs for 72 h in a final volume of 100 µl. Three by threefold PLX4032 dilutions resulted in concentrations ranging from 1 nM to 10 µM. After 72 h, the CellTiter-Glo reaction solution (Promega) was added and luminescence was measured (Tekan). Control wells with DMSO were used for normalization. $IC_{50}$ values were determined with the Compusyn software. For long-term viability assays, 800 cells in 6-well plates were treated with the indicated drugs at the $IC_{50}$ concentration. Medium was changed every 2–3 days, and colonies were fixed and stained with crystal violet after 10–15 days.

### The paper explained

**Problem**

Targeted therapies with MAPK inhibitors (MAPKi) are faced with severe problems of innate and acquired resistance in *BRAF*-mutant melanoma. No clear mutational mechanism is found in up to forty percent of resistant melanomas, indicating that transcriptomic or epigenetic alterations may underly acquired MAPKi resistance. In parallel to the acquisition of genetic mutations, melanoma cells may also adapt to the drugs through phenotype switching. Therefore, identification of non-genomic mechanisms by which melanoma cells reprogram their epigenome/transcriptome to evade MAPKi therapy may lead to the design of more efficient combination therapies.

**Results**

We herein uncovered the role of the EMT-inducing transcription factor (EMT-TF) ZEB1 as a major driver of phenotype switching in melanoma cells, providing them with a resistance to MAPKi. We showed that high ZEB1 expression levels were associated with inherent resistance to BRAF inhibitors (BRAFi) in *BRAF*^*V600*-mutated cell lines and tumors. Moreover, ZEB1 expression level was increased in melanoma cell lines with acquired resistance to BRAFi and in biopsies from patients relapsing while under BRAFi treatment. We further demonstrated that *ZEB1* overexpression in melanoma cell lines triggered the emergence of resistance to MAPKi by promoting the reversible conversion of a MITF^high/p75^low differentiated state into a MITF^low/p75^high stem-like and tumorigenic state. Consequently, the inhibition of ZEB1 sensitized naive melanoma cells to BRAFi, prevented the emergence of resistance following chronic exposure to BRAFi *in vitro*, and induced cell death in resistant melanoma cells.

**Impact**

Our work shows that mutated *BRAF* melanoma patients with high levels of ZEB1 expression may not benefit from MAPKi treatment and that ZEB1 could serve as a predictive marker in order to stratify *BRAF*-mutated melanoma into MAPKi-sensitive and MAPKi-resistant subgroups. Moreover, a better understanding of the function of EMT-TFs in melanoma cell plasticity should pave the way for the design of novel combination therapies targeting specific EMT-TFs and the MAPK pathway in order to prevent the emergence of resistance.

## Statistical analyses

All statistical analyses were performed using GraphPad Prism 6 software and Microsoft Excel 2010, except statistical analyses of TCGA and CCLE data that were performed using the R software (version 3.1.2; the R core Team R: a language and environment for statistical computing, 2008; http://www.R-project.org). To assess the significant correlation between *ZEB1*, *MITF*, and *NGFR* expression in the CCLE and TCGA data sets, a Pearson's correlation coefficient was performed. To assess the significant associations between the level of *ZEB1* expression and *BRAF*- or *NRAS*-mutated status in TCGA data set, a Wilcoxon signed-rank test was performed. All experiments were performed at least in triplicate. Data are presented as mean ± s.d. or ± s.e.m. as specified in the figure legends. To determine significant differences between two groups, parametric data were analyzed using a two-tailed *t*-test or a Fisher's exact test. The *P*-values obtained were considered significant < 0.05.

**Expanded View** for this article is available online.

## Acknowledgements

The authors would like to thank Brigitte Manship for critical reading, Michelle Houang, Sophie Leon, and Nicolas Gadot (Anipath) for their help with immunohistochemical stainings, and Frédérique Fauvet for her help with the generation of cell lines. Our team is funded by the Ligue Nationale contre le Cancer. The work was additionally supported by the Institut National du Cancer (INCa PAIR melanoma D22044), the Fondation de France (project no. 2012 00029143), and the Société Française de Dermatologie, SFD (JC). GR and ML were supported by the LABEX DEVweCAN of the University of Lyon (ANR-10-LABX-0061) and by a fellowship from the Ligue Nationale contre le Cancer (GR). AB and MPM are recipients of fellowships from the Fondation pour la Recherche Médicale (FRM).

## Author contributions

GR and MAM performed and analyzed most *in vitro* and *in vivo* studies. ML and RBar contributed to *in vitro* experiments and JC contributed to *in vivo* experiments in mice. SD, AB, MPM, GT, and LT provided human samples and clinical data and analyzed data from the human sample experiments. AF and LD performed the pathological examination of the stainings. RMP and ET performed statistical analyses of the TCGA and CCLE data. SM, MB, and CL provided human biopsies and helped to interpret data. SA helped to interpret data. CB and RBal provided cell lines and helped to interpret data. SD, AP, and JC conceived the project, designed experiments, and interpreted data. JC supervised the whole project and wrote the manuscript.

## Conflict of interest

The authors declare that they have no conflict of interest.

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
