## [Review Process File · EMBO Molecular Medicine]

ZEB1-mediated melanoma cell plasticity enhances resistance to MAPK inhibitors

Geoffrey Richard, Stéphane Dalle, Marie-Ambre Monet, Maud Ligier, Amélie Boespflug, Roxane M. Pommier, Arnaud de la Fouchardière, Marie Perier-Muzet, Lauriane Depaepe, Romain Barnault, Garance Tondeur, Stéphane Ansieau, Emilie Thomas, Corine Bertolotto, Robert Ballotti, Samia Mourah, Maxime Battistella, Céleste Lebbé, Luc Thomas, Alain Puisieux and Julie Caramel

Corresponding author: Alain Puisieux and Julie Caramel, Cancer Research Center of Lyon

Review timeline:

Submission date:	21 October 2015
Editorial Decision:	12 November 2015
Revision received:	23 May 2016
Editorial Decision:	22 June 2016
Revision received:	20 July 2016
Editorial Decision:	27 July 2016
Revision received:	09 August 2016
Accepted:	11 August 2016

Transaction Report:

Editor: Roberto Buccione

1st Editorial Decision

12 November 2015

Thank you for the submission of your manuscript to EMBO Molecular Medicine. We have now heard back from the three Reviewers whom we asked to evaluate your manuscript.

Although the very expert Reviewers I contacted agree on the potential interest of the manuscript, the issues they raise are many, of a fundamental nature and mostly overlapping. I will not dwell into much detail, but I would like to highlight the main points.

Reviewer 1 expresses two main and important concerns. On one hand s/he notes that there are several internal inconsistencies and discrepancies with the published literature. The other important point is that the data provided do not support the main conclusions. These issues are essentially a leitmotif throughout the other reviewers' comments too.

Reviewer 2 notes, in addition to some of the points made by the previous reviewer, that the analysis of more human samples is required to consolidate the medical relevance of your work. S/he also mentions that the use of cell lines is not always consistent and cross-controls are not always available and lists a number of other relevant points.

Reviewer 3 is especially appreciative of the potential importance and impact of your work (and we agree) and notes that it would be a significant step forward after your recent Cancer Cell paper. However, s/he is also quite reserved in terms of suitability for publication. His/her concerns are not so much at variance with those from Reviewers 1 and 2, but s/he is quite adamant that, as mentioned above, the clinical samples are insufficient, the A375 cell setting might not be suited and that a PDX model would have been more appropriate. This reviewer also lists many other items of concern.

Other general comments shared by the reviewers include the impression that the manuscript is not well-organized and sometimes difficult to follow and less than optimal referencing of others' work.

In conclusion, while publication of the paper cannot be considered at this stage, given the potential interest of your findings and after internal discussion, we have decided to give you the opportunity to address the criticisms.

We are thus prepared to consider a substantially revised submission, with the understanding that the Reviewers' concerns must be addressed with additional experimental data where appropriate and that acceptance of the manuscript will entail a second round of review.

If you do not have the required data available at least in part, to address the above, this might entail a significant amount of time, additional work and experimentation and might be technically challenging. On the other hand, it is clear that the Reviewers converge on the main weaknesses of your manuscript in its current form. I would therefore understand if you chose to rather seek publication elsewhere at this stage. Should you do so, and although we hope not, we would welcome a message to this effect. The overall aim is to significantly upgrade the relevance and conclusiveness of the dataset, which of course is of paramount importance for our title. I understand of course that to perform de novo animal experimentation might be especially time consuming, but these appear to be important. You are welcome to discuss specific points if you wish by contacting me via email.

Please note that it is EMBO Molecular Medicine policy to allow a single round of revision only and that, therefore, acceptance or rejection of the manuscript will depend on the completeness of your responses included in the next, final version of the manuscript.

EMBO Molecular Medicine now requires a complete author checklist (<http://embomolmed.embopress.org/authorguide#editorial3>) to be submitted with all revised manuscripts. Provision of the author checklist is mandatory at revision stage; The checklist is designed to enhance and standardize reporting of key information in research papers and to support reanalysis and repetition of experiments by the community. The list covers key information for figure panels and captions and focuses on statistics, the reporting of reagents, animal models and human subject-derived data, as well as guidance to optimise data accessibility. This checklist especially relevant in this case given the issues raised with respect to statistical treatment and animal numbers.

As you know, EMBO Molecular Medicine has a "scooping protection" policy, whereby similar findings that are published by others during review or revision are not a criterion for rejection. However, I do ask you to get in touch with us after three months if you have not completed your revision, to update us on the status. Please also contact us as soon as possible if similar work is published elsewhere.

***** Reviewer's comments *****

Referee #1 (Comments on Novelty/Model System):

Unless I missed some crucial information on the cells lines, there are inconsistencies that affect the conclusions of the manuscript and that preclude publication

Referee #1 (Remarks):

Zeb1 is an EMT-TF (Epithelial to Mesenchymal Transition Transcription Factor). These factors, key in embryonic development, have also been implicated in different steps of cancer initiation and progression. In the manuscript under consideration, Richard and colleagues used publicly available data from melanoma cell lines, patient samples, and a xenograft model to investigate the role of ZEB 1 in intrinsic and acquired resistance to MAPK pathway inhibitors in melanoma.

General comments

1.- The results are potentially interesting, but the manuscript is not very well written, figure legends lack information, the study contains some important inconsistencies and some of the conclusions are not well substantiated by the data.

2.- The authors show a significant inverse correlation between high levels of ZEB1 mRNA and sensitivity to the BRAFi, PLX4720 in a panel of cell lines from the cancer cell line encyclopedia (Figure 1). They also show that melanoma samples from patients that are intrinsically resistant to vemurafenib have high levels of ZEB1. In addition, higher ZEB1 levels were usually accompanied by a downregulation of MITF levels, and this low MITF levels had already been related to intrinsic resistance to MAPK pathway inhibitors. These results would indicate that high levels of ZEB1 would promote resistance to BRAFi. However, both in the in vitro and in vivo experiments, A375 and SKMEL5 BRAF mutant melanoma cells, two of the cell lines with a lower IC50 for PLX4720 (Supplementary Table 1), are used as a model of ZEB1^{High}/MITF^{Low} melanoma cells. Therefore, these cell lines despite showing high levels of ZEB1 are very sensitive to the BRAF inhibitor, which is inconsistent with the claims the authors are making. In addition, in page 10, last paragraph- the authors state that the ZEB1^{High}/MITF^{Low} cells A375 and SKMEL5 are more resistant to PLX4032 than 501MEL cells (IC50=80 nm) and that this cell line is ZEB1^{low}/MITF^{high}. However (i) the dose response and IC50 for PLX4032 in 501MEL cells is not shown (ii) ZEB1 and MITF levels are not shown (iii) 501MEL cells were previously shown to be resistant to PLX4032 with an IC50: 450nm) (Halaban et al., Pigment Cell and Melanoma Research, 2010). Unless this reviewer is missing some crucial information, these results do not support the conclusions reached by the authors.

3.- Figure 1 and 2. The immunostainings performed in patient samples need to be shown at higher magnifications, as it is very difficult to visualize whether the staining corresponds to either melanoma or stromal cells.

4.- Page 12, Line 16. In Fig 3 and Supplemental Fig 4, the authors compare results from ZEB1^{High}/MITF^{Low} cell lines and 501MEL regarding p75 surface expression. However, in Fig3E only results for A375 and SKMEL5 are shown. Besides, in order to show that ZEB1 promotes "stemness", they need to show both potential p75 upregulation and cell surface expression upon ZEB1 ectopic expression in 501MEL or another cell line with low endogenous ZEB 1 levels.

5.- Fig 4F. A375 xenografts are a well established model to investigate melanoma sensitivity to different inhibitors including PLX4032 (For instance, Yang H et al., Cancer Research 2010). Therefore, it is surprising that vemurafenib has only a very mild effect on the growth of the A375 xenografts. This needs to be discussed. In addition, in Materials and Methods, it is stated that tumours grew up to 1.5 cm in diameter. Does the graph reflect the volume corresponding to this size? Which was the vehicle used? How was the dose selected?

6.- Fig 4. In page 14 (data not shown) the text reads that Zeb1 knock-down does not affect A375 or SKMEL5 proliferation, but Zeb1 is known to attenuate cell proliferation by directly repressing Cyclin D transcription during EMT (Mejlvang et al, Mol Biol Cell 2007). This needs to be discussed.

In addition, there are some other specific points that need further attention.

1. The panels in the main figures are randomly placed; they do not follow the text flow or any alphabetical order. Other figures are not sufficiently explained, i.e. In Fig 1, what is it tumour from patient ESP? The figure legend for Supplementary Table 1 should be more explanatory, what are the units for the IC50?

2. Figure 3D; The authors should discuss why they think ZEB1 levels are decreasing in cells that ectopically express ZEB1 upon PLX4032 treatment

3. There are some inaccuracies in the text that need to be corrected. Page 14, line 19: the authors use melanoma cell lines. Thus, they are already transformed and therefore, they cannot use them to demonstrate that ZEB1 is required to transform cells. Page16 line 3: Vemurafenib IC50 in RPMI 17951 cells is described in Supplementary Table 1, but that is for PLX4720. A statement clarifying

the differences between vemurafenib, PLX4032 and PLX4720 should be included.

Referee #2 (Comments on Novelty/Model System):

The overall technical quality of the ms (experimental design, quality of most figures) is high but is rather deficient in statistical analyses. The novelty of the ms is high based on the lack of efficient therapies to malignant melanoma and, more importantly, the innate or acquired resistance to targeted therapies. The medical impact of the ms in its present form is considered as medium based on the small size of the resistant tumor sample. However, this impact could rise if the number of melanoma tumors with innate or acquired resistance could be increased.

The model system (gain and loss-of function studies in different melanoma cell lines; and public dataset analyses of melanomas) is adequate.

Referee #2 (Remarks):

In this ms., Richard and coauthors report that ZEB1 is associated with inherent resistance to MAPK inhibitors, being upregulated as well after acquired resistance. They propose that ZEB1 favors stemness, as shown by the increase in several melanoma initiating cell markers, and cell transformation while promotes adaptive resistance to BRAF inhibitors *in vitro*. Besides, ZEB1 depletion has the opposite effects: stemness markers decrease, clonogenic ability is reduced and xenografted tumors are smaller upon ZEB1 depletion. Moreover, the authors claim that ZEB1-depletion minimizes the appearance of resistance to MAPK inhibitors and ZEB1 silencing in already resistant cells promotes cell death. In summary, the authors argue that ZEB1 is a major player in melanoma due to its role in regulating cell plasticity which would favor stemness, tumorigenicity and resistance to MAPK inhibition.

Although the ms is well written, results and figures have remarkable quality and most results are backed by the study of melanoma human samples as well as *in vitro* experiments, the major conclusion drawn by the authors are not fully sustained by the data shown herein. One important concern relates to the size of the sample of melanoma tumors with innate or acquired resistance analyzed in the ms. Moreover, additional functional experiments should clarify the ZEB1/MITF relation in innate or acquired resistance.

MAJOR POINTS

1. First of all, it is not easy to follow some of the results in the figures shown, the panels are not properly labeled, sometimes the cell line is not even mentioned and several conclusions are only shown partially within the figures while some figures cover different experiments making it difficult to easily focus on the result mentioned. I suggest a further effort in the nomenclature of panel figures, specifying within the figure the cell line used for each experiment and try to make them more explicit. Numbering of panels in the same order as they are commented in the text should also help to ease the reading of the ms.
2. Discussion within the results regarding MITF-ZEB1 crosstalk is not straightforwardly followed and the long text does not clarify neither the results the authors show (Fig 2C) nor their conclusion regarding MITF levels. It could be useful to present their results regarding MITF IHC in BRAF mutated melanomas in the context of the results shown in Figure 1B. The same applies to mRNA data on MITF expression regarding IC50 values for BRAFi and MEKi (Suppl Fig. 1) that could be more easily followed if presented in main Fig. 1.
3. The authors state that 35% of primary resistant tumors harbor high levels of ZEB1 or TWIST1 (Fig. 1B), this figure accounts for about 5 out of 15 non-responder patients. Therefore, it would be interesting to show the % corresponding to only high ZEB1 levels since TWIST does not seem to have such a prominent role in the resistance to inhibitors or the authors fail to demonstrate it. On the other hand, the number of patient tumors used to assess the higher expression of ZEB1 upon acquired resistance (3 out of 5) is too small to draw a conclusion. It is highly recommended to extend the sample size and to present ZEB1 and MITF IHC for more than a single patient (Fig. 2E). Importantly, the BRAF mutant (or wt) and the ALK/Wnt5A status of those patients should be presented to ascertain whether apparent ZEB1 mediated resistance is dependent or independent of those pathways.
4. The above point is directly related to the ZEB1/MITF relation that remains not completely explored along the ms. First, the inverse ZEB1/MITF correlation is only analyzed at protein level in

a few melanoma cell lines (Fig. 2B), and this study should be extended to, at least, some of the BRAF mutant cell lines with different sensitivity to MAPKi (shown in Suppl Table 1). Second, the fact that MITF levels varied significantly between different resistant cell lines (Fig. 2D) all of them with high ZEB1 levels (Fig. 1D) makes difficult to reconcile with the switch from MITF-high/p75-low to MITF-low/p75-high phenotype mediated by ZEB1 OE and associated to intrinsic resistance to MAPKi. Resistance to MAPKi in MITF low (or negative)/ AXL+/Wnt5A+ tumors have been shown to be independent of BRAF mutant status. Therefore, it should be important to determine the status of those pathways in the different melanoma cell lines studied

5. Regarding the claim ZEB1 favoring stemness (Figs. 3 and 4) the authors might overinterpreting their results since most of them are only shown for 1 cell line instead of the three they work with, and it is not clear why so few results are shown regarding 501MEL cell line. In this cell line clearer results might be expected upon ZEB1 OE due to the low endogenous levels. In addition, which is the effect, if any, of ZEB1 OE in tumorigenic properties of the three cell lines? The authors only analyze colony-forming capacity that it is an indication but not formal demonstration of in vivo increased tumorigenicity by ZEB1 OE as the authors conclude.

6. In the opposite ZEB1 loss-of-function studies performed on A375 cells, the authors analyze the tumorigenic capacity after inducible ZEB1 KD (Fig. 4F) and conclude that ZEB1 KD decreased tumor growth. Indeed, the data shown indicate that ZEB1 KD increased tumor latency but not tumor growth once surviving cells are able to start growing as demonstrated by the similar slope of the graphs from control and shZEB1 cells. Indeed, these data will suggest that ZEB1 KD is affecting tumor initiation. Importantly, control of efficient ZEB1 silencing along the in vivo experiment is required to draw definitive conclusion on the action of ZEB1 on initiation and/or tumor growth. On the other hand, the lack of apparent effect of ZEB1 KD upon vemarufenib treatment in the tumorigenic properties is also difficult to reconcile with the in vitro colony assays (Fig. 4E).

7. The undifferentiated state promoted by ZEB1 (and associated to a drug resistant stem-like phenotype) requires further characterization. Established melanoma differentiation markers, apart from MITF, should be analyzed at protein level after gain and loss of function models. Analyses of EMT markers (E-cadherin) and EMT-TFs described by the authors as expressed mainly in melanocytes (Snail2 and ZEB2) should also be included. These data could reinforce the author's hypothesis as presented in Fig. 5C.

8. The authors do not show two related results although one of their major conclusions is based on them: ZEB1 OE promotes early resistance to chronic treatment with PLX4032 (page 13, last paragraph); and ZEB1 silencing prevents the appearance of resistance (page 15, last paragraph).

9. Statistical significance is missing in most quantified data. This is particularly important for Fig. 1D; Fig. 3B, C, E, F; FIG. 4B, C, E; and Suppl Fig. 4A, B; and Suppl. Fig. 6.

MINOR POINTS

Fig. 2C: Is not clear that there are more colonies in A375-ZEB1 cell line, they appear to be only bigger than in control cells

Fig. 2E and 4C: is p75 increase in A375-ZEB1 cells and decrease in SKMEL-shZEB1 cells, respectively, significant?

Fig. 4E: These pictures are not clear to draw author's conclusion and data presented in adjacent graph.

Fig. 4B: Data presented in images for colony-forming assays require proper quantification and statistical analyses.

Suppl. Figs 4A, 6 and 7: Images from colony-forming assays are of low quality not allowing appreciate differences between controls an experimental situations

Referee #3 (Comments on Novelty/Model System):

The main conclusions are essentially based on work performed with only two established melanoma cell lines.

The work on clinical materials is performed with sample sizes that are too small to draw any firm conclusions.

The work in vivo is performed using one melanoma cell line -A375 - which did not show response to the BRAFi. This model is not appropriate to test their hypothesis.

Instead the authors should use a series of short-term cultures for the in vitro work and PDX for the in vivo work.

Experiments with clinical samples should be performed on many more cases.

The role of ZEB1 in drug resistance has already been proposed - but arguably not studied as extensively as in this manuscript.

The literature is not reviewed and cited properly.

Referee #3 (Remarks):

In this manuscript Richard et al. make an attempt to establish an association between ZEB1 levels and intrinsic/acquired resistance of melanoma cells to MAPK-inhibitors. They show that ZEB1 over-expression facilitates the emergence of resistance to BRAFV600E-inhibitors by promoting a reversible transition towards MITF low and p75high stem-like and tumorigenic phenotype. Conversely they report that ZEB1 silencing decreases the tumorigenic potential of melanoma cells and increases their sensitivity to BRAFV600E-inhibitors.

A key question in the field is: can resistance to MAPK-targeting therapeutics be acquired through (reversible) epigenomic, as opposed to genomic, alterations? Recent evidence by Hugo et al., Cell 2015 provides support for this possibility. Understanding the mechanisms by which melanoma cells can reprogram their epigenome/transcriptome to evade therapy is therefore of great interest both on a biological and clinical point of view.

The role of ZEB1 in drug resistance has already been proposed by the authors themselves (in their excellent Cancer Cell paper) and by others - but arguably it has not been investigated as directly and extensively as in this manuscript.

In this context, this study is novel, interesting and particularly timely.

Unfortunately, in its present form, the enclosed study falls short in convincing this particular referee that ZEB1-mediated transcriptome reprogramming could contribute to drug resistance.

-The main conclusions are essentially based on work performed with only two established melanoma cell lines.

The rationale for the choice of particular cell lines is also sometimes unclear. In Figure 3 the authors assess the impact of ZEB1 overexpression on the sensitivity to BRAFV600E-inhibitors. They used A375 and SKMEL5, two cell lines which express high ZEB1 (see Figure 2B). Why not use at least one additional, ZEB1-negative, cell line such as 501Mel?

-The work on clinical materials is performed with sample sizes that are too small to draw any firm conclusions. Is there a significant association between high ZEB1 alone (and not ZEB1+TWIST1) and primary resistance to treatment (Fig1E). The rationale for including TWIST1 in this group is totally unclear in light of the data presented in relation to Figure 1A. Fig. 1E says 3 out of 5 patients show the expected result. This is obviously such a small sample size that no conclusion can be drawn from this experiment.

-The work in vivo is performed using xenograft with one melanoma cell line -A375 - which did not show a clear response to the BRAFi. This model is not appropriate to test the author's hypothesis. Instead, the authors should use a series (n>5) of short-term cultures for the in vitro work and PDX for the in vivo work. Experiments with clinical samples should be performed using many more cases...

The authors claim that they demonstrate that ZEB1-mediated resistance is a direct consequence of its ability to promote reprogramming towards a MITFlow, p75high stem-like phenotype. However the epistatic relationship has not been tested. Their conclusion is solely based on correlative observations. The relevance of varying levels of p75 for the observed phenotypes should be tested experimentally.

As opposed to irreversibility of genetic alterations, one feature of epigenomic adaptation to treatment is its reversibility. The authors did not attempt to test this possibility. This is to my opinion

an interesting hypothesis that the authors should set out to test using inducible/reversible ZEB1 KD model systems.

The authors actually claim (see highlights) that ZEB1 promote reversible conversions between a differentiated and stem-like state. This has not been addressed at all. In fact neither the reversibility nor the ability of ZEB1 to promote a full conversion from a differentiated to stem-like phenotype have been tested.

The literature is not properly reviewed and cited. Papers from the Garraway group (Cancer Discovery paper) and Peeper (Muller et al., Nat Commun) groups have shown that MITF levels determine the sensitivity of melanoma cells to BRAF/MEK-inhibitors. Their papers should be cited. The implications of phenotype switching model to resistance to MAPK-therapy have already been reviewed in Kemper et al. 2014.

Few additional specific points:

-Figure 3A -why show only 2 out of 3 cell lines? why not show MITF protein levels (same question for Fig4A)? Why not show p75 mRNA levels?

Does ZEB1 overexpression affect cell behavior? Cell proliferation rate?

Are exogenous ZEB1 levels comparable to what is seen in resistant tumors?

-3F and 5B should be supplemented by careful evaluation of the EC50 corresponding to the different treatments.

Quantification of the nb of colonies Fig5B should also be shown.

-How was the specificity of the ZEB1 and TWIST1 antibodies used for IHC tested?

-Page 8 -a dramatic increase ...? I would not call a two-fold increase a dramatic increase.

-The order of the panels does not fit with the order in the text. Ex. Go from 3B to 3E, then to 3C, ...

-Figure 4F, Actine is written in French.

-Fig4E - the authors conclude that ZEB1 is required for the transformation of melanoma cells. This is not what the experiment shows. The experiment shows that ZEB1 is required for the maintenance of melanoma growth *in vitro*.

-Fig4E -there is nothing to be seen in the PLX-treated cells only (sh-Control)- it is unclear how ZEB1 KD could aggravate this phenotype.

1st Revision - authors' response

23 May 2016

Detailed responses to the referees comments:

We thank the three referees for their useful comments that we have addressed as described below. For clarity, initial comments of the reviewers are indicated in black and our answers are in blue.

Referee #1 (Comments on Novelty/Model System):

Unless I missed some crucial information on the cells lines, there are inconsistencies that affect the conclusions of the manuscript and that preclude publication

Our study highlights a phenotype-mediated MAPKi resistance mechanism, with a critical role of ZEB1. Our conclusion that high levels of ZEB1 are associated to resistance to MAPKi is especially strengthened by the analyses of human samples that reinforce the correlation observed *in vitro* in cell lines. Nevertheless, as explained below, ZEB1 is not the only factor influencing resistance since several other mechanisms have been described previously.

Referee #1 (Remarks):

Zeb1 is an EMT-TF (Epithelial to Mesenchymal Transition Transcription Factor). These factors, key in embryonic development, have also been implicated in different steps of cancer initiation and progression. In the manuscript under consideration, Richard and colleagues used publicly available data from melanoma cell lines, patient samples, and a xenograft model to investigate the role of ZEB 1 in intrinsic and acquired resistance to MAPK pathway inhibitors in melanoma.

General comments

1.- The results are potentially interesting, but the manuscript is not very well written, figure legends lack information, the study contains some important inconsistencies and some of the conclusions are not well substantiated by the data.

In order to take these comments into consideration, the order of the manuscript has been significantly modified and figure legends were completed to improve the clarity of our manuscript.

Moreover, additional experiments both in short-term cultures and in a patient-derived xenograft model were included to reinforce our data.

2.- The authors show a significant inverse correlation between high levels of ZEB1 mRNA and sensitivity to the BRAFi, PLX4720 in a panel of cell lines from the cancer cell line encyclopedia (Figure 1). They also show that melanoma samples from patients that are intrinsically resistant to vemurafenib have high levels of ZEB1. In addition, higher ZEB1 levels were usually accompanied by a downregulation of MITF levels, and this low MITF levels had already been related to intrinsic resistance to MAPK pathway inhibitors. These results would indicate that high levels of ZEB1 would promote resistance to BRAFi. However, both in the *in vitro* and *in vivo* experiments, A375 and SKMEL5 BRAF mutant melanoma cells, two of the cell lines with a lower IC50 for PLX4720 (Supplementary Table 1), are used as a model of ZEB1^{High}/MITF^{Low} melanoma cells. Therefore, these cell lines despite showing high levels of ZEB1 are very sensitive to the BRAF inhibitor, which is inconsistent with the claims the authors are making.

In order to confirm the significant inverse correlation between high levels of ZEB1 and sensitivity to MAPKi evidenced *in silico* in the cell lines from the cancer cell line encyclopedia (Figure 1D), we analyzed a larger set of cell lines *in vitro*, and demonstrated the significance of the inverse correlation:

These results are now included in the text (p6-7) and in new figure 1B, C and E:

“We then confirmed these results by conducting quantitative PCR (Q-PCR) and Western blot analyses in a panel of 14 *BRAF*^{V600}-mutated human melanoma cell lines, including two short-term cultures established from patients with melanomas displaying similar mutations (GLO and C-09.10) (Fig 1B and C). We observed an inverse correlation between the levels of ZEB1 and those of ZEB2 and MITF, while TWIST1 protein levels were generally high in all of these cell lines and were not correlated with MITF (Fig 1B and C). [...]

We then validated these findings in our panel of *BRAF*^{V600}-mutated melanoma cell lines, by determining the IC50 of PLX4032. To do so, we performed ATP assays, in which melanoma cells were treated with this drug, at a dose ranging from 1 nM to 10 μM, for 72 h. The IC50 for PLX4032 was generally higher in the ZEB1^{high}/MITF^{low} cell lines, compared to the ZEB1^{low}/MITF^{high} cell lines (Fig 1E). Overall, these *in silico* and *in vitro* data demonstrate that cell lines intrinsically resistant to MAPKi exhibit a ZEB1^{high}/MITF^{low} profile.”

Although significant, we agree with this referee that the correlation between high ZEB1 levels and resistance to MAPKi in a panel of cell lines *in vitro* is not absolute. It is also not the case for the correlation between low MITF levels and intrinsic MAPKi resistance although previously nicely shown by the Garraway and Peeper labs (Konieczkowski et al., 2014; Muller et al., 2014). Namely, A375 and SKMEL5 cell lines exhibit high levels of ZEB1 and low levels of MITF while being rather sensitive to BRAFi. However, this should not be considered as inconsistent with our model:

- Indeed, functional experiments demonstrate that modulation of ZEB1 levels modulates the sensitivity to MAPKi in these two models. The level of ZEB1 expression is further increased in these cell lines upon acquisition of resistance to BRAFi, as demonstrated in the A375-R

and SKMEL5-R resistant models (Figure 3B), and the ectopic expression of ZEB1 triggers the emergence of resistance (Figure 4, figures EV2 and Appendix S2).

- Moreover, the association between high levels of ZEB1 expression and intrinsic resistance to treatment was shown to be highly significant in human samples, thus reinforcing the conclusions drawn from the *in vitro* work (Figure 2D).

- Finally, as previously mentioned, we do not argue that ZEB1 is the only factor influencing sensitivity to MAPKi treatment and we do not exclude the putative co-occurrence of other mechanisms of resistance that could account for the differential sensitivity to treatment in established cell lines *in vitro*.

In addition, in page 10, last paragraph- the authors state that the ZEB1^{High}/MITF^{Low} cells A375 and SKMEL5 are more resistant to PLX4032 than 501MEL cells (IC₅₀=80 nm) and that this cell line is ZEB1^{low}/MITF^{high}.

However (i) the dose response and IC₅₀ for PLX4032 in 501MEL cells is not shown

The IC₅₀ values for the whole panel of cell lines have now been calculated and are shown in figure 1E. They are consistent with the data from the CCLE (Table EV1).

(ii) ZEB1 and MITF levels are not shown

ZEB1 and MITF expression levels were already shown in 501MEL and a few other cell lines in the previous figure 2B. This is now shown in a larger panel of cell lines, at both the protein and mRNA levels, in the current figure 1B and C, see the Results section (p6).

(iii) 501MEL cells were previously shown to be resistant to PLX4032 with an IC₅₀: 450nm (Halaban et al., Pigment Cell and Melanoma Research, 2010). Unless this reviewer is missing some crucial information, these results do not support the conclusions reached by the authors.

The expected major genetic alterations were verified by NGS sequencing in the 501MEL cell line. We reproducibly found that 501MEL cells are sensitive to PLX4032 with an IC₅₀ below 100 nM, similarly to other ZEB1^{low}/MITF^{high} cell lines (WM9, MALM3M, M4BE...). However we do agree that A375 and SKMEL5 cell lines are not significantly more resistant to BRAFi.

We conclude that “The IC₅₀ for PLX4032 was generally higher in the ZEB1^{high}/MITF^{low} cell lines, compared to the ZEB1^{low}/MITF^{high} cell lines (Fig 1E).”

3.- Figure 1 and 2. The immunostainings performed in patient samples need to be shown at higher magnifications, as it is very difficult to visualize whether the staining corresponds to either melanoma or stromal cells.

As requested by this reviewer we have now shown pictures at a higher magnification (Figure 2 and 3). Stromal and endothelial cells (indicated with arrows, Fig 2E and Fig 3E) are indeed positive for ZEB1, but can be easily distinguished from melanoma cells by the pathologist (Dr A. de la Fouchardière, French national referent in melanomas) who analyzed the stainings.

4.- Page 12, Line 16. In Fig 3 and Supplemental Fig 4, the authors compare results from ZEB1^{High}/MITF^{Low} cell lines and 501MEL regarding p75 surface expression. However, in Fig3E only results for A375 and SKMEL5 are shown. Besides, in order to show that ZEB1 promotes "stemness", they need to show both potential p75 upregulation and cell surface expression upon ZEB1 ectopic expression in 501MEL or another cell line with low endogenous ZEB 1 levels.

We agree with this reviewer that it is important to show the consequences of the ectopic expression of ZEB1 in at least one ZEB1^{low} cell line in the main figures. We therefore analyzed the putative induction of p75 following the ectopic expression of ZEB1 in various ZEB1^{low} models. p75 could not be induced by ZEB1 in various ZEB1^{low} established cell lines (501MEL, MALM3M, M4BE), but was readily activated in the two *BRAF*^{V600} patient-derived short-term cultures, C-09.10 and GLO. C-09.10 cells were thus presented in the main figure 5, while 501MEL and GLO are included in expanded view figures EV2 and EV3.

This is now indicated in the Results section (p14):

“We then investigated the consequences of ectopically expressing *ZEB1* in ZEB1^{low}/MITF^{high} cells. Surprisingly, in the ZEB1^{low}/MITF^{high} cell lines, such as 501MEL, *ZEB1* overexpression was not

sufficient to promote p75 expression, even upon PLX4032 treatment (Fig EV2A-C). However, *ZEB1* expression increased the clonogenic growth of *ZEB1*^{low} 501MEL and, whereas PLX4032 treatment drastically inhibited the growth of control cells in soft agar, *ZEB1*-overexpressing cells were less sensitive to the BRAFi in this assay (Fig EV2D), suggesting that *ZEB1* can promote resistance in this model without p75 induction.

Finally, to investigate the function of *ZEB1* in physiological models with low levels of *ZEB1* expression, the EMT inducer was ectopically expressed in two *BRAF*^{V600} patient-derived short-term cultures, C-09.10 cells and GLO (Fig 5 and Fig EV3). *ZEB1* ectopic expression in C-09.10 cells led to a significant decrease in *MITF* levels and increase in p75 levels (Fig 5A, B and D). Similarly in GLO cells, *ZEB1* ectopic expression was shown to promote the conversion into a p75^{high} state, which was potentiated upon PLX4032 treatment (Fig EV3). *ZEB1*-induced phenotype switching was associated to an increased capacity to form colonies in soft agar (Fig 5C) and to resistance to BRAFi as assessed by a clonogenic assay in the presence of PLX4032 (Fig 5E). Results in these two short-term culture models therefore validated the conclusions obtained in established *ZEB1*^{high} cell lines, in a *ZEB1*^{low} context.”

This is now commented in the Discussion section (p20):

“*ZEB1* ectopic expression was sufficient to increase p75 levels in *ZEB1*^{high} established cell lines (A375, SKMEL5), and *ZEB1*^{low} patient-derived short-term cultures (C09.10 and GLO), and this effect was potentiated upon BRAFi treatment. However, in *ZEB1*^{low}/*MITF*^{high} established melanoma cell lines, such as 501MEL, *ZEB1* overexpression was not sufficient to promote p75 expression, even upon PLX4032 treatment. This may be due to epigenetic modifications acquired during the establishment of these cell lines that may block the promoter in a closed chromatin configuration.”

5.- Fig 4F. A375 xenografts are a well established model to investigate melanoma sensitivity to different inhibitors including PLX4032 (For instance, Yang H et al., Cancer Research 2010). Therefore, it is surprising that vemurafenib has only a very mild effect on the growth of the A375 xenografts. This needs to be discussed. In addition, in Materials and Methods, it is stated that tumours grew up to 1.5 cm in diameter. Does the graph reflect the volume corresponding to this size? Which was the vehicle used? How was the dose selected?

We agree with this reviewer that the effect of vemurafenib on the growth of A375 xenografts, presented in the initial version of the manuscript, was not as strong as expected. We speculated that this was due to the mode of administration by IP injection. Therefore, we reproduced the experiments and opted for a *per os* route of administration, 50 mg/kg, once a day. By doing so, the effect of vemurafenib on A375/SKMEL5 sensitive cells was very significant (expanded view Figure EV4D).

Additional information has now been included in the Material and methods section, “Mouse injections” (p25):

“BRAF was inhibited by orally administering vemurafenib (50 mg/kg) or vehicle (DMSO/PBS) daily for 2-5 weeks.”

Moreover, in order to demonstrate the synergy of BRAFi and *ZEB1* inhibition, we also included another xenograft model from resistant cells, namely short-term culture ESP, since the *in vitro* effect of *ZEB1* depletion was stronger in resistant compared to sensitive cells. Data are now shown in figure 7F and G (as detailed in the response to Reviewer#2, comment n°6 and Reviewer#3, comment n°3).

6.- Fig 4. In page 14 (data not shown) the text reads that *Zeb1* knock-down does not affect A375 or SKMEL5 proliferation, but *Zeb1* is known to attenuate cell proliferation by directly repressing Cyclin D transcription during EMT (Mejlvang et al, Mol Biol Cell 2007). This needs to be discussed.

ZEB2 (SIP1) ectopic expression in epidermoid A431 cells was shown to attenuate proliferation by direct repression of Cyclin D1 (Mejlvang et al, 2007). However, in our melanoma models, the ectopic expression of *ZEB1* does not affect cell proliferation under 2D growing conditions (proliferation curves were performed). This reflects cell-type specific functions of *ZEB1* and *ZEB2* in melanoma compared to carcinoma, as we previously reported (Caramel et al, 2013; Puisieux et al

2014). ZEB2 ectopic expression in melanoma cell lines was indeed associated to decreased tumorigenic capacity.

Discussion about proliferation and referencing of (Mejlvang et al, 2007) is now included (p22):

“Of note, while the ectopic expression of *ZEB1/ZEB2* has been associated to attenuated cell proliferation in carcinoma models (Mejlvang *et al.*, 2007), we did not observe any proliferation defect upon *ZEB1* expression in melanoma cells, further highlighting cell-type specific functions of *ZEB1* and *ZEB2* in melanoma compared to carcinoma, as we previously reported (Caramel *et al.*, 2013; Puisieux *et al.*, 2014).”

In addition, there are some other specific points that need further attention.

1. The panels in the main figures are randomly placed; they do not follow the text flow or any alphabetical order. Other figures are not sufficiently explained, i.e. In Fig 1, what is it tumour from patient ESP? The figure legend for Supplementary Table 1 should be more explanatory, what are the units for the IC50?

We thank this reviewer for these suggestions. Order of panels and legends were modified accordingly and legends of figures and tables were completed, as requested by reviewer#1.

2. Figure 3D; The authors should discuss why they think *ZEB1* levels are decreasing in cells that ectopically express *ZEB1* upon PLX4032 treatment.

This issue is now addressed in the Discussion section (p20-21):

“We previously showed that *ZEB1* expression levels are not only regulated at the transcriptional level by the *FRA1* transcription factor, but also at the post-translational level (Caramel *et al.*, 2013). This may explain why *ZEB1* levels are still decreasing in cells that ectopically express *ZEB1* upon PLX4032 treatment.”

3. There are some inaccuracies in the text that need to be corrected. Page 14, line 19: the authors use melanoma cell lines. Thus, they are already transformed and therefore, they cannot use them to demonstrate that *ZEB1* is required to transform cells.

We agree with the reviewer and modified the text accordingly (p12). Moreover, “xenograft experiments in *nude* mice were performed with control or *ZEB1*-overexpressing A375 cells and revealed a significant increase in tumor growth in the latter case (Fig 4D).”

Page16 line 3: Vemurafenib IC50 in RPMI 17951 cells is described in Supplementary Table 1, but that is for PLX4720. A statement clarifying the differences between vemurafenib, PLX4032 and PLX4720 should be included.

Vemurafenib/PLX4032 and PLX4720 are analogs of the same molecule. PLX4032 was chosen (over PLX4720) for further clinical development because of its better pharmacokinetic properties shown in the preclinical studies *in vivo*. The range of IC50 values for PLX4720 in the CCL6 are highly similar to those we determined with PLX4032 in our cell lines.

This is now included in the legends of Figure 1 and Table EV1.

Referee #2 (Comments on Novelty/Model System):

The overall technical quality of the ms (experimental design, quality of most figures) is high but is rather deficient in statistical analyses.

We thank this reviewer for the positive comment on the quality of the manuscript. Statistical analyses have now been performed as appropriate for the different types of experiments (Q-PCR analyses, quantification of colonies, etc...).

The novelty of the ms is high based on the lack of efficient therapies to malignant melanoma and, more importantly, the innate or acquired resistance to targeted therapies. The medical impact of the ms in its present form is considered as medium based on the small size of the resistant tumor sample.

However, this impact could rise if the number of melanoma tumors with innate or acquired resistance could be increased.

As suggested by this reviewer, we have now significantly increased the size of the human sample cohort as indicated below.

The model system (gain and loss-of function studies in different melanoma cell lines; and public dataset analyses of melanomas) is adequate.

We thank this reviewer for recognizing the diversity of the models used in our study.

Referee #2 (Remarks):

In this ms., Richard and coauthors report that ZEB1 is associated with inherent resistance to MAPK inhibitors, being upregulated as well after acquired resistance. They propose that ZEB1 favors stemness, as shown by the increase in several melanoma initiating cell markers, and cell transformation while promotes adaptative resistance to BRAF inhibitors in vitro. Besides, ZEB1 depletion has the opposite effects: stemness markers decrease, clonogenic ability is reduced and xenografted tumors are smaller upon ZEB1 depletion. Moreover, the authors claim that ZEB1-depletion minimizes the appearance of resistance to MAPK inhibitors and ZEB1 silencing in already resistant cells promotes cell death. In summary, the authors argue that ZEB1 is a major player in melanoma due to its role in regulating cell plasticity which would favor stemness, tumorigenicity and resistance to MAPK inhibition. Although the ms is well written, results and figures have remarkable quality and most results are backed by the study of melanoma human samples as well as in vitro experiments, the major conclusion drawn by the authors are not fully sustained by the data shown herein. One important concern relates to the size of the sample of melanoma tumors with innate or acquired resistance analyzed in the ms. Moreover, additional functional experiments should clarify the ZEB1/MITF relation in innate or acquired resistance.

As requested by this reviewer, we have both increased the size of the human sample cohort and clarified the relationship between ZEB1/MITF by changing the order of the presentation as indicated below.

MAJOR POINTS

1. First of all, it is not easy to follow some of the results in the figures shown, the panels are not properly labeled, sometimes the cell line is not even mentioned and several conclusions are only shown partially within the figures while some figures cover different experiments making it difficult to easily focus on the result mentioned. I suggest a further effort in the nomenclature of panel figures, specifying within the figure the cell line used for each experiment and try to make them more explicit. Numbering of panels in the same order as they are commented in the text should also help to ease the reading of the ms.

As already suggested by Reviewer#1, we have reorganized the figures and completed the legends. We strongly believe that these changes have significantly increased the clarity of the manuscript. We acknowledge the reviewers for their constructive comments.

2. Discussion within the results regarding MITF-ZEB1 crosstalk is not straightforwardly followed and the long text does not clarify neither the results the authors show (Fig 2C) nor their conclusion regarding MITF levels. It could be useful to present their results regarding MITF IHC in BRAF mutated melanomas in the context of the results shown in Figure 1B. The same applies to mRNA data on MITF expression regarding IC50 values for BRAFi and MEKi (Suppl F1g. 1) that could be more easily followed if presented in main Fig. 1.

As suggested by this reviewer we modified the order and now present the association between the ZEB1/MITF ratio and resistance to treatment:

- in new figure 1: *in silico* and *in vitro* in melanoma cell lines;
- in new figure 2: in human samples.

3. The authors state that 35% of primary resistant tumors harbor high levels of ZEB1 or TWIST1 (Fig. 1B), this figure accounts for about 5 out of 15 non-responder patients. Therefore, it would be interesting to show the % corresponding to only high ZEB1 levels since TWIST1 does not seem to have such a prominent role in the resistance to inhibitors or the authors fail to demonstrate it.

The number of human samples obtained from the pre-treatment cohort was increased to 70 samples, by including 40 new patients (new figure 2D) as detailed in the text (p8):

“To determine whether the levels of ZEB1 and MITF were predictive of the patients’ response to MAPKi, we performed immunohistochemical staining for ZEB1, MITF but also TWIST1 on a cohort of 70 human *BRAF*^{V600}-melanoma samples from patients whose response to the treatment was known. Thirty patients presented a primary resistance (initial non-responders), and 40 were initial responders but relapsed during their treatment with MAPKi (developing acquired resistance). Sixteen of those patients received combined treatment with the MEK inhibitor cobimetinib.”

The analyses confirmed that high levels of ZEB1 expression alone are significantly associated with primary resistance to MAPKi as mentioned in the Results section (p9):

“Interestingly, most ZEB1^{high} melanoma samples were in the primary resistance group (Fig 2D). Thirty percent of primary resistant melanomas exhibited high levels of endogenous ZEB1, compared to only 7.5% of the initially responding tumors ($p=2.27E-2$) (Fig 2D). Moreover, the samples with high levels of ZEB1 exhibited low levels of MITF (Patient 1, before treatment, Fig 2E).”

Although the expression of TWIST1 in cell lines is not associated with MAPKi resistance, high levels of TWIST1 expression in patients are significantly associated with resistance as mentioned in the Results (p9):

“Interestingly, a significant proportion of ZEB1-negative melanoma samples from patients with primary resistance displayed strong TWIST1 staining. Collectively, 50% of primary resistant melanomas exhibited a strong staining for ZEB1 and/or TWIST1. Thus, initial high levels of endogenous ZEB1 and/or TWIST1 were associated with primary resistance to treatment ($p=3E-4$) (Fig 2D).”

We agree that TWIST1 appears to play a secondary role in MAPKi resistance compared to ZEB1, as reported in the Discussion section (p22):

“While our study was mainly focused on ZEB1, TWIST1 is also frequently activated in melanoma. However, the ectopic expression of *TWIST1* did not significantly stimulate the colony formation capacity of melanoma cell lines (Appendix Figure S7), suggesting that ZEB1 is a stronger oncogenic factor than TWIST1 in this tumor type. Nevertheless, if the ectopic expression of *TWIST1* is unable to confer resistance to PLX4032 in ZEB1^{high} cell lines (A375/SKMEL5), our results indicate that TWIST1 can drive resistance in a ZEB1^{low} context (501MEL cells, Fig EV2). When combined with the observation that a significant number of primary resistant ZEB1-negative melanomas exhibit high levels of TWIST1 expression, these results suggest that ZEB1 is the main driver of BRAFi resistance, but that TWIST1 may complement ZEB1 when this factor is not activated, and may thus constitute a potential therapeutic target in a subset of melanomas.”

On the other hand, the number of patient tumors used to assess the higher expression of ZEB1 upon acquired resistance (3 out of 5) is too small to draw a conclusion. It is highly recommended to extent the sample size and to present ZEB1 and MITF IHC for more than a single patient (Fig. 2E).

Regarding matched sample pairs, before and after treatment in patients, we already presented 8 sample pairs (3 for primary resistant samples and 5 upon acquired resistance) in the initial version of the manuscript. These biopsies are rare and this material is difficult to obtain. Most studies in the field usually show no more than 5 pairs (see for example, Muller et al., Nat Commun. 2014). However, we managed to further increase this sample size, by analyzing 4 more pairs, for ZEB1 and MITF staining, representative pictures of which are now shown in the new figure 3E.

As indicated in the Results section (p10-11):

“In 5 out of 9 matched pre-treatment and post-relapse samples pairs of acquired resistance, ZEB1 proteins either appeared or their levels had further increased (Fig 3E). In the samples obtained from an initially responding patient, both ZEB1 and TWIST1 were present at low levels and then increased significantly in the relapsed tumor after treatment, in terms of both the intensity and percentage of positive cells, which changed from 10% to 80% (Patient 4, Fig 3E and Fig EV1B). We observed a decrease in MITF levels after treatment in 2/8 patients (some clones of Patient 1, Fig

2E), while these levels were maintained (Patient 2, Fig 3E) or increased (Patients 3 and 4, Fig 3E) in most melanoma relapse samples after vemurafenib treatment.”

Importantly, the BRAF mutant (or wt) and the ALK/Wnt5A status of those patients should be presented to ascertain whether apparent ZEB1 mediated resistance is dependent or independent of those pathways.

BRAF^{V600} mutation was validated in all human samples before treatment.

Invasive MITF^{low} cells with high levels of expression of WNT5A or of the Receptor Tyrosine Kinase AXL were previously shown to be more resistant to MAPKi (Anastas et al., 2014;Konieczkowski et al., 2014;Muller et al., 2014). Levels of AXL and WNT5A expression were thus analyzed by Western blot or Q-PCR in our patient-derived BRAFi-resistant short-term cultures. While WNT5A could not be detected, high levels of AXL were detected in GOKA and ESP, as well as in A375-R in SKMEL5-R resistant cells (Appendix Figure S6). AXL is a known target of EMT in carcinoma (Sayan et al., 2012). However, neither AXL nor WNT5A levels were increased upon *ZEB1* ectopic expression in A375 and SKMEL5 cells (see below, response to comment n°7), suggesting that ZEB1 function in resistance may be independent of these pathways. AXL and ZEB1 pathways may thus be two mechanisms of resistance that may co-occur but not be redundant.

This is now indicated in the Discussion section (p20).

4. The above point is directly related to the ZEB1/MITF relation that remains not completely explored along the ms. First, the inverse ZEB1/MITF correlation is only analyzed at protein level in a few melanoma cell lines (Fig. 2B), and this study should be extended to, at least, some of the BRAF mutant cell lines with different sensitivity to MAPKi (shown in Suppl Table 1).

This question was already addressed in the response to Reviewer#1, comment n°2.

This issue highlights the specific roles of ZEB1 and the importance of considering the ZEB1/ZEB2 ratio.

Second, the fact that MITF levels varied significantly between different resistant cell lines (Fig, 2D) all of them with high ZEB1 levels (Fig. 1D) makes difficult to reconcile with the switch from MITF-high/p75-low to MITF-low/p75-high phenotype mediated by ZEB1 OE and associated to intrinsic resistance to MAPKi.

Our results are consistent with data previously published by other groups: a MITF^{low} phenotype is associated with intrinsic resistance to MAPKi (Konieczkowski et al., 2014;Muller et al, 2014), but MITF high or low levels of expression have been found in acquired resistance samples (Johannessen et al., 2013;Muller et al., 2014) highlighting the dual function of this factor, known as the MITF rheostat model (Hoek and Goding, 2010).

As indicated in the Discussion section (p21):

“The role of MITF is complex, since both high and low levels of MITF can be found in cells with acquired resistance, even within the same tumor in different clones, suggesting that MITF may or may not be required for the acquisition and maintenance of resistance to MAPKi (Wellbrock and Arozarena, 2015)”.

Resistance to MAPKi in MITF low (or negative)/ AXL+/Wnt5A+ tumors have been shown to be independent of BRAF mutant status. Therefore, it should be important to determine the status of those pathways in the different melanoma cell lines studied

AXL and WNT5A could not be detected at the protein level in most of the sensitive cell lines analyzed in this study, in contrast with resistant models. Moreover, neither AXL nor WNT5A levels were increased upon *ZEB1* ectopic expression in A375 and SKMEL5 cells.

This is now included in the Results section (p12):

“Moreover, expression levels of some invasion markers, such as *Vimentin*, *SPARC* or *MMPI*, were slightly induced in *ZEB1*-overexpressing cells, while *AXL* and *WNT5A* levels were not modified (Appendix Figure S3B).”

5. Regarding the claim ZEB1 favoring stemness (Figs. 3 and 4) the authors might over-interpreting their results since most of them are only shown for 1 cell line instead of the three they work with,

and it is not clear why so few results are shown regarding 501MEL cell line. In this cell line clearer results might be expected upon ZEB1 OE due to the low endogenous levels.

This question was already addressed in the response to Reviewer#1, comment n°4.

In addition, which is the effect, if any, of ZEB1 OE in tumorigenic properties of the three cell lines? The authors only analyze colony-forming capacity that it is an indication but not formal demonstration of *in vivo* increased tumorigenicity by ZEB1 OE as the authors conclude.

We agree with this reviewer that colony-forming capacity in soft agar was not sufficient to draw a definitive conclusion. We thus performed *in vivo* xenograft experiments in *nude* mice, that demonstrate the increased tumorigenic potential of ZEB1 over-expressing cells, as detailed in the Results section (p12):

“Xenograft experiments in *nude* mice were performed with control or *ZEB1*-overexpressing A375 cells and revealed a significant increase in tumor growth in the latter case (Fig 4D).”

6. In the opposite ZEB1 loss-of-function studies performed on A375 cells, the authors analyze the tumorigenic capacity after inducible ZEB1 KD (Fig. 4F) and conclude that ZEB1 KD decreased tumor growth. Indeed, the data shown indicate that ZEB1 KD increased tumor latency but not tumor growth once surviving cells are able to start growing as demonstrated by the similar slope of the graphs from control and shZEB1 cells. Indeed, these data will suggest that ZEB1 KD is affecting tumor initiation.

The experiment that was shown with inducible ZEB1-shRNA *in vivo* did not allow to address the question of the impact of ZEB1 knock-down on tumor initiation, since ZEB1 knock-down was induced by IPTG treatment when tumors had already grown to a volume of 100mm³.

In order to demonstrate the impact of ZEB1 knock-down on tumor initiation, additional xenograft experiments have now been performed with A375 cells expressing a constitutive ZEB1-shRNA and demonstrated the absence of tumor initiation, as detailed in the Results section (p15):

“We then examined whether the presence of ZEB1 was a requirement for the growth of malignant melanoma cells *in vivo*. As expected from previous data in melan-a and B16F10 murine cells (Caramel et al., 2013; Dou et al., 2014), ZEB1 knock-down in A375 cells prevented tumor initiation in *nude* mice (Fig 6E), clearly demonstrating that ZEB1 is required for the tumorigenic capacity of melanoma cells.”

The experiment performed with inducible shRNA then allows to demonstrate the impact on tumor growth of established tumors (p15):

“We thus used IPTG-inducible ZEB1-shRNA to evaluate the impact of ZEB1 knock-down on tumor shrinkage in established tumors. Two ZEB1-shRNAs that consistently reduced the levels of endogenous ZEB1 protein upon IPTG treatment *in vitro* were used (Fig 6F). The A375 cells infected with the IPTG-inducible control- or ZEB1-shRNA were injected subcutaneously into *nude* mice. When tumors reached a diameter of 5 mm, the mice were given IPTG in their drinking water. ZEB1 knock-down led to a drastic decrease in tumor growth, confirming the potent anti-tumor effect of ZEB1 inhibition (Fig 6F).”

Importantly, control of efficient ZEB1 silencing along the *in vivo* experiment is required to draw definitive conclusion on the action of ZEB1 on initiation and/or tumor growth. On the other hand, the lack of apparent effect of ZEB1 KD upon vemurafenib treatment in the tumorigenic properties is also difficult to reconcile with the *in vitro* colony assays (Fig. 4E).

Efficient ZEB1 silencing in the tumor upon IPTG treatment was controlled by directly assessing ZEB1 protein levels in the tumors by western blot and immunostaining for ZEB1 (Figure 7G) (p17).

Since the consequences of ZEB1 knock-down *in vitro* were stronger in BRAFi-resistant cells compared to sensitive cells (Figure 7B, D and E) with an induction of cell death only in resistant cells, we decided to establish another xenograft model from vemurafenib-resistant patient-derived short-term culture, namely ESP cells. While the synergy between ZEB1 knock-down and BRAFi treatment was not significant in sensitive SKMEL5 cells (Fig EV4D), this was the case in the resistant ESP models as indicated in the Results section (p17):

“Control or ZEB1-shRNA ESP cells were xenografted into *nude* mice and orally treated with IPTG +/- vemurafenib. While vemurafenib did not affect tumor growth of control resistant cells, ZEB1

inhibition alone or in combination with vemurafenib led to a significant decrease in tumor growth (Fig 7F).”

7. The undifferentiated state promoted by ZEB1 (and associated to a drug resistant stem-like phenotype) requires further characterization. Established melanoma differentiation markers, apart from MITF, should be analyzed at protein level after gain and loss of function models. Analyses of EMT markers (E-cadherin) and EMT-TFs described by the authors as expressed mainly in melanocytes (Snail2 and ZEB2) should also be included. These data could reinforce the author’s hypothesis as presented in Fig. 5C.

To better characterize the undifferentiated and differentiated state, apart from the regulation of MITF, p75, ABCB5 and JARID1B that were already included, we now analyzed the expression of additional differentiation markers (Tyrosinase (TYR), ZEB2, and E-Cadherin), and EMT/invasion markers (Vimentin, SPARC, MMP1, AXL, WNT5A,) upon *ZEB1* ectopic expression or knock-down.

- We now show modulation of ZEB2 levels by western blot and Q-PCR upon *ZEB1* expression in Appendix Figure S3A and upon *ZEB1* knock-down in Appendix Figure S5, as detailed in the Results section:

p11: “ZEB1 ectopic expression triggered [...] upregulation of ZEB2 (Appendix Figure S3A).”

p15: “An increase in [...] ZEB2 and E-Cadherin expression levels [...] were observed upon *ZEB1* knock-down in A375 cells (Appendix Figure S5).”

- E-cadherin is not detected in most melanoma cell lines but its levels are increased upon *ZEB1* knock-down as we previously reported (Caramel et al, 2013). This is now shown in Appendix Figure S5.

- While the differentiation marker TYR (Tyrosinase) is only detected at very low levels in A375 and SKMEL5 cells, it is completely lost upon *ZEB1* expression. Since quantification of such low levels was not possible, these results were not shown.

- Finally, “expression levels of some invasion markers, such as *Vimentin*, *SPARC* or *MMP1*, were slightly induced in *ZEB1*-overexpressing cells, while *AXL* and *WNT5A* levels were not modified (Appendix Figure S3B).” (p12 of the Results section).

8. The authors do not show two related results although one of their major conclusions is based on them: *ZEB1* OE promotes early resistance to chronic treatment with PLX4032 (page 13, last paragraph); and *ZEB1* silencing prevents the appearance of resistance (page 15, last paragraph).

These results have now been included in the main figures 4I and 7C, respectively.

9. Statistical significance is missing in most quantified data. This is particularly important for Fig. 1D; Fig. 3B, C, E, F; FIG. 4B, C, E; and Suppl Fig. 4A, B; and Suppl. Fig. 6.

The statistical significance (p value) has now been added to the figures, as required by Reviewer#2.

MINOR POINTS

Fig. 2C: Is not clear that there are more colonies in A375-ZEB1 cell line, they appear to be only bigger than in control cells

Quantification of the colonies clearly revealed that the number of colonies was increased following *ZEB1* overexpression. We agree that colonies growing from *ZEB1*-expressing A375 cells are also

bigger than those obtained from control cells, suggesting an additional effect on cell proliferation in these 3D conditions.

Fig. 2E and 4C: is p75 increase in A375-ZEB1 cells and decrease in SKMEL-shZEB1 cells, respectively, significant?

A Fisher exact test confirmed the significance of p75 differential expression assessed by FACS analyses in ZEB1-expressing *versus* Control cells.

Fig. 4E: These pictures are not clear to draw author's conclusion and data presented in adjacent graph.

Pictures of colonies have now been shown at a higher magnification in order to be consistent with quantification shown in adjacent graphs (new Figure 7B).

Fig. 4B: Data presented in images for colony-forming assays require proper quantification and statistical analyses.

Quantification of the number of colonies obtained in the clonogenic assays performed with the resistant cell models has now been shown in adjacent graphs to the pictures, together with statistical analyses (new Figure 7E).

Suppl. Figs 4A, 6 and 7: Images from colony-forming assays are of low quality not allowing appreciate differences between controls and experimental situations

The quality of the pictures in new figures (EV2, EV4, and S7) has now been improved, as required by Reviewer#2.

Referee #3 (Comments on Novelty/Model System):

The main conclusions are essentially based on work performed with only two established melanoma cell lines.

The main conclusions were based on work performed in two established BRAFi-sensitive melanoma cell lines (A375, SKMEL5) but also in 501MEL, RPMI7951, as well as in established BRAFi-resistant cell lines (A375-R, SKMEL5-R) and in two patient-derived BRAFi-resistant short-term cultures (GOKA, ESP). These results have now been confirmed in additional BRAFi-sensitive patient-derived short-term cultures models.

The work on clinical materials is performed with sample sizes that are too small to draw any firm conclusions.

We have now significantly increased the human sample size and validated our conclusions (Figures 2 and 3).

The work *in vivo* is performed using one melanoma cell line -A375 - which did not show response to the BRAFi. This model is not appropriate to test their hypothesis.

As indicated by the two other reviewers, the A375 cells are an appropriate model to test the sensitivity to BRAFi *in vivo*. However, we agree that BRAFi efficacy was not as strong as expected in this model in our experimental conditions; we therefore changed the route of BRAFi administration for IP to PO, which has resulted in a significant effect (Figure EV4 and Figure 7).

Instead the authors should use a series of short-term cultures for the *in vitro* work and PDTX for the *in vivo* work.

We have now added two BRAFi-sensitive patient-derived short term cultures, C-09. 10 and GLO. These new data support our conclusions following the ectopic expression of ZEB1 (Figure 5 and Figure EV3). Moreover, knock-down experiments were already presented in two patient-derived BRAFi-resistant short-term cultures, GOKA and ESP.

In addition, in order to test our hypotheses in more physiological conditions *in vivo*, we developed another xenograft model, from patient-derived short-term culture (Figure 7).

Experiments with clinical samples should be performed on many more cases.

Additional clinical samples have been included, not only for the pre-treatment samples (from 30 to 70) but also for the biopsies analyzed before and after treatment (Figures 2 and 3).

The role of ZEB1 in drug resistance has already been proposed - but arguably not studied as extensively as in this manuscript.

We thank the reviewer for emphasizing the novelty of our study, which provided in depth characterization of the function of ZEB1 in MAPKi resistance both *in vitro* and *in vivo*, in human samples and xenograft models.

The literature is not reviewed and cited properly.

We are sorry, but the original articles mentioned by Reviewer#3 were already cited.

Referee #3 (Remarks):

In this manuscript Richard et al. make an attempt to establish an association between ZEB1 levels and intrinsic/acquired resistance of melanoma cells to MAPK-inhibitors. They show that ZEB1 over-expression facilitates the emergence of resistance to BRAFV600E-inhibitors by promoting a reversible transition towards MITF low and p75high stem-like and tumorigenic phenotype. Conversely they report that ZEB1 silencing decreases the tumorigenic potential of melanoma cells and increases their sensitivity to BRAFV600E-inhibitors.

A key question in the field is: can resistance to MAPK-targeting therapeutics be acquired through (reversible) epigenomic, as opposed to genomic, alterations? Recent evidence by Hugo et al., Cell 2015 provides support for this possibility. Understanding the mechanisms by which melanoma cells can reprogram their epigenome/transcriptome to evade therapy is therefore of great interest both on a biological and clinical point of view.

The role of ZEB1 in drug resistance has already been proposed by the authors themselves (in their excellent Cancer Cell paper) and by others - but arguably it has not been investigated as directly and extensively as in this manuscript.

In this context, this study is novel, interesting and particularly timely.

We thank Reviewer#3 for recognizing the impact and the novelty of our results. The role of an "EMT" invasive phenotype has indeed been suggested by other studies (WNT5A, AXL), though no investigations have so far focused on the specific function of EMT-TFs. Based on our previous data showing that ZEB1 and TWIST1 may have an antagonistic function to ZEB2, specifically in the melanocyte lineage, we now demonstrate that ZEB1 is a major driver of phenotype-mediated MAPKi resistance.

Unfortunately, in its present form, the enclosed study falls short in convincing this particular referee that ZEB1-mediated transcriptome reprogramming could contribute to drug resistance.

We have now included additional models both *in vitro* and *in vivo* (included in figure 1, 5 and 7), and integrated a larger cohort of human patients (in figures 2 and 3). We are therefore confident that our article has been significantly improved.

1- The main conclusions are essentially based on work performed with only two established melanoma cell lines. The rationale for the choice of particular cell lines is also sometimes unclear. In Figure 3 the authors assess the impact of ZEB1 overexpression on the sensitivity to BRAFV600E-inhibitors. They used A375 and SKMEL5, two cell lines which express high ZEB1 (see Figure 2B). Why not use at least one additional, ZEB1-negative, cell line such as 501Mel?

The ectopic expression of ZEB1 was already performed in 501MEL, although results were shown in former supplemental figure 4 (new Expanded view Figure EV2). We agree with this reviewer that it is important to show the consequences of the ectopic expression of ZEB1 in at least one ZEB1^{low} cell line in the main figures. We therefore performed these experiments in the two patient-derived BRAF^{V600} short term-cultures, C-09.10 and GLO, which are ZEB1^{low}/MITF^{high}. Results are now shown in the main Figure 5 and in Expanded view Figure EV3 as already detailed in the response to Referee#1 comment n°4.

2- The work on clinical materials is performed with sample sizes that are too small to draw any firm conclusions. Is there a significant association between high ZEB1 alone (and not ZEB1+TWIST1) and primary resistance to treatment (Fig1E). The rationale for including TWIST1 in this group is totally unclear in light of the data presented in relation to Figure 1A. Fig. 1E says 3 out of 5 patients show the expected result. This is obviously such a small sample size that no conclusion can be drawn from this experiment.

This question was already addressed in the response to Reviewer#2, comment n°3.

3- The work in vivo is performed using xenograft with one melanoma cell line -A375 - which did not show a clear response to the BRAFi. This model is not appropriate to test the author's hypothesis.

Instead, the authors should use a series (n>5) of short-term cultures for the in vitro work and PDTX for the in vivo work. Experiments with clinical samples should be performed using many more cases...

Although the A375 model is commonly used to test sensitivity to BRAFi *in vivo*, we agree that the efficacy of vemurafenib was not as strong as expected in our experimental conditions. We believe that this was due to the mode of administration by IP injection. Therefore we reproduced the experiments and opted for a *per os* route of administration, 50mg/kg, once a day. By doing so, the effect of vemurafenib on A375/SKMEL5 sensitive cells was very significant (Expanded view Figure EV4D).

Regarding the *in vitro* models, we have now added two BRAFi-sensitive patient-derived short term cultures, namely C-09.10 and GLO, in which we confirmed our findings following the ectopic expression of ZEB1 as detailed previously. Moreover, knock-down experiments were already presented in two patient-derived BRAFi-resistant short-term cultures, ESP and GOKA (new Figure 7).

Since the consequences of ZEB1 knock-down *in vitro* were stronger in BRAFi-resistant cells compared to sensitive cells (Figure 7B, D and E), with an induction of cell death only in resistant cells, we decided to establish another xenograft model from vemurafenib-resistant patient-derived short-term cultures, namely ESP cells. This technique is similar to patient-derived xenografts (PDXs), in which we would have had difficulties in determining the function of ZEB1, since the *in vitro* infection with shRNA-ZEB1 is required before injection into *nude* mice. While the synergy between ZEB1 knock-down and BRAFi treatment was not significant in sensitive SKMEL5 cells (Fig EV4D), this was the case in the resistant ESP models as indicated in the Results section (p17):

“Control or ZEB1-shRNA ESP cells were xenografted into *nude* mice and orally treated with IPTG +/- vemurafenib. While vemurafenib did not affect tumor growth of control resistant cells, ZEB1 inhibition alone or in combination with vemurafenib led to a significant decrease in tumor growth (Fig 7F).”

4- The authors claim that they demonstrate that ZEB1-mediated resistance is a direct consequence of its ability to promote reprogramming towards a MITFlow, p75high stem-like phenotype. However the epistatic relationship has not been tested. Their conclusion is solely based on correlative observations. The relevance of varying levels of p75 for the observed phenotypes should be tested experimentally.

We agree with his reviewer that we had not demonstrated the role of increased p75 levels in the ZEB1-mediated phenotype. In order to verify whether p75 is required for the observed phenotype, we performed knock-down experiments with siRNA directed against p75 and demonstrated that at least part of the effects mediated by ZEB1 is dependent on the increased expression of p75.

This is now detailed in the Results section (p13):

“To investigate the role of p75 in the ZEB1-mediated phenotype, siRNA experiments against p75 were performed in A375 cells (Fig 4G). The knock-down of p75 in ZEB1-overexpressing A375 cells resulted in a level of p75 equivalent to that in control cells. The knock-down of p75 induced an increase in MITF expression levels in ZEB1-expressing cells similar to that in control cells, thus suggesting that ZEB1-mediated downregulation of MITF is dependent on p75. We conclude that p75 is at least responsible for part of the effects associated with ZEB1.”

This is also reported in the Discussion section (p19-20):

“However, knock-down experiments demonstrated that p75 was only responsible for part of the effects associated with ZEB1, since the ZEB1-mediated decrease in *Tyrosinase* was not reverted after the knock-down of *p75*. Moreover, *ZEB1* expression promoted resistance to BRAFi in 501MEL cells without induction of p75, suggesting that ZEB1 can promote resistance by other mechanisms. The expression of two other markers that do not necessarily overlap with p75 (Cheli et al., 2014), namely the ABCB5 transporter and the histone demethylase JARID1B (Roesch et al., 2010; Schatton et al., 2008), were also regulated upon ZEB1 modulation. Overall, ZEB1, as a transcription factor which can act both as a transcriptional repressor or activator thanks to the binding to specific co-factors, is responsible for the modulation of a large panel of targets, including down-regulation of melanocyte differentiation markers and upregulation of melanoma initiating cell markers, that cooperate in mediating resistance to MAPKi.”

5- As opposed to irreversibility of genetic alterations, one feature of epigenomic adaptation to treatment is its reversibility. The authors did not attempt to test this possibility. This is to my opinion an interesting hypothesis that the authors should set out to test using inducible/reversible ZEB1 KD model systems.

The authors actually claim (see highlights) that ZEB1 promote reversible conversions between a differentiated and stem-like state. This has not been addressed at all. In fact neither the reversibility nor the ability of ZEB1 to promote a full conversion from a differentiated to stem-like phenotype have been tested.

The ability of ZEB1 to promote a full conversion from a differentiated to a stem-like phenotype is based on the analyses of a large series of markers, which has been further increased in the new version of the manuscript, as already mentioned in the response to Reviewer#2, comment n°7.

Moreover, the consequences of *ZEB1* gain or loss of function on the initiating/tumorigenic capacity has now been demonstrated, as mentioned in the response to Reviewer#2, comments n°5-6.

In addition, to demonstrate that ZEB1 promotes the reversible conversion between these two states, we took advantage of IPTG-inducible shRNA-ZEB1 that was previously described and used in the *in vivo* xenograft experiments.

As indicated in the Results section (p15-16):

“Finally, to demonstrate the reversibility of the ZEB1-mediated phenotype switching, the expression of *ZEB1*-shRNA was induced for 10 days (+IPTG), ZEB1 expression was then reversed by removing IPTG for 10 days (-IPTG). Upon IPTG withdrawal, levels of ZEB1, MITF and p75 returned to the basal levels in untreated cells (Fig 6G). Taken together with the results obtained following *ZEB1* overexpression, our data indicate that ZEB1 drives the reversible conversion of MITF^{high}/p75^{low} differentiated into MITF^{low}/p75^{high} stem-like/initiating phenotypes, and regulates the subsequent tumorigenic capacity of melanoma cells.”

6- The literature is not properly reviewed and cited. Papers from the Garraway group (Cancer Discovery paper) and Peeper (Muller et al., Nat Commun) groups have shown that MITF levels determine the sensitivity of melanoma cells to BRAF/MEK-inhibitors. Their papers should be cited. The implications of phenotype switching model to resistance to MAPK-therapy have already been reviewed in Kemper et al. 2014.

These two major references from the Garraway and Peeper laboratories were already cited, in the Results section (p10 of the previous version of the manuscript; p7 of the revised version):

“low MITF expression was shown to predict intrinsic resistance to MAPKi (Konieczkowski et al., 2014; Muller et al., 2014)”

as well as in the Discussion section (p18 in the previous version; p20 in the revised version):

“In support of our model, invasive MITF^{low} cells with high expression of WNT5A or of the Receptor Tyrosine Kinase AXL were recently shown to be more resistant to MAPKi (Anastas et al., 2014; Konieczkowski et al., 2014; Muller et al., 2014).”

The interesting review from Kemper et al. 2014, was not included due to space limitations, since we gave priority to original articles.

Few additional specific points:

-Figure 3A -why show only 2 out of 3 cell lines?

We now show results for ZEB1^{high} A375 cells in the main figure 4 (former figure 3) and similar results in SKMEL5 cells in Appendix Figure S2.

Results in ZEB1^{low} short-term cultures C09-10 are now shown in the main figure 5, while results in 501MEL and GLO are now shown in expanded view Figures EV2 and EV3.

why not show MITF protein levels (same question for Fig4A)? Why not show p75 mRNA levels?

Analyses of p75 mRNA expression levels are now shown for all of the models (Figures 4, 5, 6, and appendix Fig S2).

Does ZEB1 overexpression affect cell behavior? Cell proliferation rate?

ZEB1 overexpression does not affect the rate of cellular proliferation (as detailed in the response to Reviewer#1, comment n°6), but does affect migration/invasion properties as we previously reported (Caramel et al, Cancer Cell, 2013).

Are exogenous ZEB1 levels comparable to what is seen in resistant tumors?

ZEB1 levels observed upon ectopic expression are equivalent to that in resistant cell lines as evidenced by Q-PCR (3-4 fold increase in ZEB1-expressing A375 or SKMEL5 cells, compared to a 2-fold increase in the resistant models).

-3F and 5B should be supplemented by careful evaluation of the EC50 corresponding to the different treatments.

IC50 values for PLX4032 were determined in sensitive A375 cells upon ZEB1 overexpression or knock-down. However, the differences were modest after 3 days of treatment (3-fold increase or decrease respectively upon ZEB1 overexpression or knock-down), compared to the significant differences observed after 2 weeks of treatment in the clonogenic assays. These observations indicate that ZEB1 effects rely on a process of drug-induced phenotype adaptation that requires at least one week of treatment. This is consistent with the data showing that p75 induction by ZEB1 is potentiated after 10 days of treatment with PLX4032.

This is now discussed in the Discussion section (p19):

“Since IC50 values for PLX4032 were only moderately modified upon ZEB1 overexpression or knock-down after 3 days of treatment in sensitive A375 cells, this further indicates that ZEB1 effects rely on a process of drug-induced phenotype adaptation that requires at least one week of treatment. Moreover, cell death was observed upon ZEB1 knock-down in A375-R cells even in the absence of PLX4032 treatment (Fig 7D), indicating that these resistant cells are addicted to ZEB1.”

Quantification of the nb of colonies Fig5B should also be shown.

Quantification of the number of colonies obtained in the clonogenic assays performed with the resistant cell models has now been shown in adjacent graphs to the pictures, together with statistical analyses (new Figure 7E).

-How was the specificity of the ZEB1 and TWIST1 antibodies used for IHC tested?

The specificity of ZEB1 (H102, Santa-Cruz) and TWIST1 (Twist2C1a, Abcam) antibodies used for IHC staining of human samples was previously validated by our expert pathologists (Caramel et al, Cancer Cell, 2013). Positive controls included stromal cells for ZEB1 and TWIST1 as well as endothelial cells for ZEB1 (Fig 2E, Fig 3E), while epithelial cells served as negative controls on the same slides.

-Page 8 -a dramatic increase ...? I would not call a two-fold increase a dramatic increase.

We agree and replaced “dramatic” by “strong” in this sentence, in the Results section (p10):

“The resistant cells displayed a strong increase in their levels of ZEB1 protein and mRNA compared to their parental counterparts (Fig 3B and C).”

However, we would like to emphasize that a two-fold increase in the level of ZEB1 mRNA expression results in a very significant increase in protein level (as shown in Figure 3B).

-The order of the panels does not fit with the order in the text. Ex. Go from 3B to 3E, then to 3C, ...

As previously requested by the two other reviewers, we have now changed the order of the panels in order to fit with the order in the text.

-Figure 4F, Actine is written in French.

This was modified as requested by Reviewer#3 in new Figure 6F. We apologize for the mistake.

-Fig4E - the authors conclude that ZEB1 is required for the transformation of melanoma cells. This is not what the experiment shows. The experiment shows that ZEB1 is required for the maintenance of melanoma growth *in vitro*.

We agree with this referee and modified the text accordingly. Moreover, we further demonstrated the effect of ZEB1 knock-down on tumorigenic growth *in vivo* (Results section, p15):

“ZEB1 knock-down in A375 cells prevented tumor initiation in *nude* mice (Fig 6E), clearly demonstrating that ZEB1 is required for the tumorigenic capacity of melanoma cells.”

-Fig4E -there is nothing to be seen in the PLX-treated cells only (sh-Control)- it is unclear how ZEB1 KD could aggravate this phenotype.

Pictures are now shown at higher magnification, as requested by Reviewer#3 (new Figure 7B).

2nd Editorial Decision

22 June 2016

Thank you for the submission of your revised manuscript to EMBO Molecular Medicine. We have now heard back from the three Reviewers whom we asked to evaluate your manuscript.

I apologise for delay in getting back to you. We experienced difficulties in obtaining the reviewer evaluations in a timely manner. In addition to this, your case required further discussion with my colleagues on the way forward.

You will see that, while reviewer 3 is now satisfied that his/her concerns have been adequately addressed, reviewers 1 and 2 instead remain reserved and point to number of important pending issues that would require adequate action. I am especially worried about the concerns on the IC50 experiments for PLX, which reviewer 2 finds to be inadequate and in want of much better experimental evidence. The same reviewer, as does reviewer 1, also notes conflicting evidence on the role of p75.

Although we would normally not allow a second significant revision, based on the reviewer evaluations and our discussions I am prepared in this case, to give you the opportunity to improve your manuscript by responding to each point and providing additional experimental evidence where necessary as mentioned above. Depending on the completeness of your response, I may be able to make an editorial decision on your next, final version.

As you know, EMBO Molecular Medicine has a "scooping protection" policy, whereby similar findings that are published by others during review or revision are not a criterion for rejection. However, I do ask you to get in touch with us after three months if you have not completed your revision, to update us on the status. Please also contact us as soon as possible if similar work is published elsewhere.

I would also encourage you to provide better quality images in general. In fact, we note excessive pixilation/blurring when magnifying your figures and also excess contrasting in some instances.

I look forward to seeing a revised form of your manuscript as soon as possible.

***** Reviewer's comments *****

Referee #1 (Comments on Novelty/Model System):

Still believe that the A375 and SKMEL5 are not the best models to study in this context, as in spite

of expressing high Zeb1 they have a low EC50 for PLX. Nevertheless, the authors have very much improved the manuscript and have toned down the conclusions and commented on when some inconsistencies or not perfect fits with the main conclusions were apparent.

Referee #1 (Remarks):

The manuscript is very much improved after revision. The flow is much better, the order of panels in Figures and the description of all data shown are much clearer. Also, the inclusion of additional clinical samples and experiments help to reinforce the conclusions.

Minor comments

- In the patient-derived short-term cultures it would be desirable to see an increase in EC50 after Zeb1 overexpression
- Fig. 6F--it is not clear that the result reflects "a drastic decrease in tumor growth"--in addition, p value is very high.
- Fig. 7B-- The panel does not reflect a decrease in the number of colonies, but rather that they are smaller
- Coming back to the models used, it is surprising that the authors have not carried out more experiments in a cell line high for Zeb and more resistant to PLX. The cell line used in Fig. 7E could be an example--however, they have only used it to test its clonogenicity --what about MIFT or p75 levels?

Referee #2 (Comments on Novelty/Model System):

The ms presents novel data of potential translational interest in the melanoma area. The technical and overall scientific quality has been greatly improved in the revised versions. The model systems (cell lines and xenografts) have been also improved as well as provide more confident data from patient samples and derived short cultures

Referee #2 (Remarks):

Richard and coauthors have performed relevant experiments to back up their hypothesis regarding the major role played by ZEB1 in resistance to MAPK inhibitors, both intrinsic and acquired, in melanoma cell lines and patients. The main conclusions are now supported by improved results obtained after increasing the number of human samples analyzed, the complementary experiments performed in nude mice and the general restructuring of the results presented.

The authors have appropriately answered to the major concerns raised and the manuscript has improved accordingly. In order to recommend the acceptance for publication of the current manuscript, I would suggest that the authors address a few concerns beforehand.

1. The Supplementary Methods included in the Appendix only show primers for Q-PCR assays but not additional methodology; therefore I couldn't review them nor learn what is the ATP assay used for IC50 determination (page 7. Lines 78 from bottom).
2. Regarding the IC50 for PLX4032 two major concerns arise. First, an n = 2 to determine the IC50 for PLX4032 shown in Fig. 1E does not seem adequate. Additionally, results displayed in Fig. 3A are surprising as well since seem to come from n = 1 experiment.
3. Moreover, the data presented in EV Table 1 should be properly discussed since it is not obvious the claimed inversed relationship between resistance to PLX4032 and high ZEB1 levels. The authors should explain the rationale behind the colors chosen for the table cells, as well as from which IC50 is considered sensitivity or resistance, or the basis for considering low or high ZEB1 expression.
4. Regarding p75 and due to the relevant role assigned by the authors in melanoma stemness, there seems to be highly discrepant and opposite basal p75 protein levels in the same cell line displaying the same levels of ZEB1 in two different experiments (A375 cells in Fig. 4A vs Fig. 6A). This discrepancy should be explained.

MINOR POINTS

Scale bar should be in μm and not in M .

Pictures in Fig. 3D are not informative.; they should be removed or shown as Appendix information

Referee #3 (Comments on Novelty/Model System):

PDX is a well-accepted/valid model system

Referee #3 (Remarks):

The authors have adequately addressed all my concerns/criticisms

2nd Revision - authors' response

20 July 2016

Detailed responses to the referees comments:

We thank referee 3 for his/her kind reply and referees 1 and 2 for their useful comments that we have addressed as described below. For clarity, initial comments of the reviewers are indicated in black and our answers are in blue.

Referee #1 (Comments on Novelty/Model System):

Still believe that the A375 and SKMEL5 are not the best models to study in this context, as in spite of expressing high Zeb1 they have a low EC50 for PLX. Nevertheless, the authors have very much improved the manuscript and have toned down the conclusions and commented on when some inconsistencies or not perfect fits with the main conclusions were apparent.

Referee #1 (Remarks):

The manuscript is very much improved after revision. The flow is much better, the order of panels in Figures and the description of all data shown are much clearer. Also, the inclusion of additional clinical samples and experiments help to reinforce the conclusions.

Minor comments

- In the patient-derived short-term cultures it would be desirable to see an increase in EC50 after Zeb1 overexpression

In the patient-derived short-term cultures C-09.10 and GLO, IC50 for PLX4032 had already been calculated after ZEB1 expression. A slight increase (although not significant) in IC50 after 3 days of treatment is observed upon ZEB1 expression in C-09.10 and GLO cells. These results are now shown in Appendix Figure S7. The effects of ZEB1 expression are more impressive in the clonogenic assays after 2 weeks of treatment, as already discussed for A375 cells. This is now indicated in the discussion section (p19):

“Since IC50 values for PLX4032 were only moderately modified upon ZEB1 overexpression after 3 days of treatment in sensitive A375, C-09.10 or GLO cells (Appendix Figure S7), this further indicates that ZEB1 effects rely on a process of drug-induced phenotype adaptation that requires at least one week of treatment.”

- Fig. 6F--it is not clear that the result reflects "a drastic decrease in tumor growth"--in addition, p value is very high.

We agree with this referee that the decrease in tumor growth upon inducible knock-down of *ZEB1* (Figure 6F) is less impressive than upon constitutive knock-down (Figure 6E) and thus replaced “drastic” by “significant” (p15).

- Fig. 7B-- The panel does not reflect a decrease in the number of colonies, but rather that they are smaller

The quantitative analyses shown in Fig 7B indicate a significant decrease in the number of colonies. In addition, we agree with this referee that the size of the colonies is also diminished. This observation has now been included in the text (p16):

“This was associated with a concomitant decrease in the size of the colonies.”

- Coming back to the models used, it is surprising that the authors have not carried out more experiments in a cell line high for Zeb and more resistant to PLX. The cell line used in Fig. 7E could be an example--however, they have only used it to test its clonogenicity --what about MITF or p75 levels?

We have now analyzed the expression levels of p75 and MITF in the ZEB1^{high} vemurafenib-resistant ESP cells upon ZEB1 knock-down (Fig 7F).

“Moreover, p75 expression was decreased and MITF expression was increased upon ZEB1 knock-down in ESP resistant cells (Fig 7F).”

Referee #2 (Comments on Novelty/Model System):

The ms presents novel data of potential translational interest in the melanoma area.

The technical and overall scientific quality has been greatly improved in the revised versions. The model systems (cell lines and xenografts) have been also improved as well as provide more confident data from patient samples and derived short cultures

Referee #2 (Remarks):

Richard and coauthors have performed relevant experiments to back up their hypothesis regarding the major role played by ZEB1 in resistance to MAPK inhibitors, both intrinsic and acquired, in melanoma cell lines and patients. The main conclusions are now supported by improved results obtained after increasing the number of human samples analyzed, the complementary experiments performed in nude mice and the general restructuring of the results presented.

The authors have appropriately answered to the major concerns raised and the manuscript has improved accordingly. In order to recommend the acceptance for publication of the current manuscript, I would suggest that the authors address a few concerns beforehand.

1. The Supplementary Methods included in the Appendix only show primers for Q-PCR assays but not additional methodology; therefore I couldn't review them nor learn what is the ATP assay used for IC50 determination (page 7. Lines 78 from bottom).

As mentioned in the main Material and Methods section (p29, Viability assays), the CellTiter-Glo kit from Promega was used for determination of IC50:

“Viability assays

For short term viability assays, the CellTiter-Glo Luminescent Cell Viability Assay (ATP assay) (Promega) was used, based on quantitation of the ATP present, which signals the presence of metabolically active cells.”

2. Regarding the IC50 for PLX4032 two major concerns arise. First, an n = 2 to determine the IC50 for PLX4032 shown in Fig. 1E does not seem adequate. Additionally, results displayed in Fig. 3A are surprising as well since seem to come from n = 1 experiment.

We now show the mean IC50 determined from n=3 experiments in Fig 1E and Fig 3A.

3. Moreover, the data presented in EV Table 1 should be properly discussed since it is not obvious the claimed inversed relationship between resistance to PLX4032 and high ZEB1 levels. The authors should explain the rationale behind the colors chosen for the table cells, as well as from which IC50 is considered sensitivity or resistance, or the basis for considering low or high ZEB1 expression.

As presented in Figure 1D (Tukey box plot), the level of *ZEB1* mRNA is inversely correlated with the sensitivity to the BRAFi PLX4720 with a significant p value ($p=2E-4$, $n=28$). The source data from the CCLE are now represented in Appendix Figure S1 (previous EV Table 1, in order to keep it in color, since for production purposes tables should be in black and white only), and the legend has now been completed to explain the color code:

“Cell lines are ranked according to their PLX4032 IC50. Color code: IC50<5 μ M: green, IC50>5 μ M: red ; *MITF*, *ZEB1*, *TWIST1* and *ZEB2* expression levels: <7: green, >7: red.”

4. Regarding p75 and due to the relevant role assigned by the authors in melanoma stemness, there seems to be highly discrepant and opposite basal p75 protein levels in the same cell line displaying the same levels of ZEB1 in two different experiments (A375 cells in Fig. 4A vs Fig. 6A). This discrepancy should be explained.

The apparent differences in p75 protein levels in the A375 cell line in Fig. 4A vs Fig. 6A was due to different time exposure of the western-blot in the two figures. We have now shown in addition to a high exposure, a low exposure of the same western-blot in Fig. 6A, showing equivalent levels of p75 in control cells than in Fig 4A. Moreover source data of these western-blot are now included in appendix.

MINOR POINTS

Scale bar should be in μ m and not in μ M.

We apologize for the mistake; this modification has now been made in all figures.

Pictures in Fig. 3D are not informative; they should be removed or shown as Appendix information

Pictures in Fig. 3D are now shown in Expanded view Figure 1C.

Referee #3 (Comments on Novelty/Model System):

PDX is a well-accepted/valid model system

Referee #3 (Remarks):

The authors have adequately addressed all my concerns/criticisms

We thank referee 3 for his/her kind review and referees 1 and 2 for their useful comments.

3rd Editorial Decision

27 July 2016

Thank you for the submission of your revised manuscript to EMBO Molecular Medicine. The remaining reviewer is now globally supportive and I am pleased to inform you that we will be able to accept your manuscript pending a few final amendments.

I have noticed a few discrepancies/mixups in your source data file that require resolution before we can move forward. Once fixed, please upload separate source data files for each figure

Also, please provide the synopsis figure as a standalone file.

Please submit your revised manuscript within two weeks. I look forward to seeing a revised final form of your manuscript as soon as possible.

***** Reviewer's comments *****

The manuscript is suitable for publication

Corresponding Author Name: CARMEL Julie
 Journal Submitted to: EMBO Molecular Medicine
 Manuscript Number: EMM-2015-05971